# Comprehensive protocol for mixed reality visualization and navigation using 3D Slicer

**Ziyu Qi**[1]*, **Christopher Nimsky**[1,2], **Miriam H. A. Bopp**[1,2]*

**1** Department of Neurosurgery, University of Marburg, Baldingerstrasse, Marburg, Germany, **2** Center for Mind, Brain and Behavior (CMBB), Marburg, Germany

* qizi@staff.uni-marburg.de (ZQ); bauermi@med.uni-marburg.de (MHAB)

## Abstract

Advancements in mixed reality (MxR) technology have significantly enhanced neurosurgical navigation by improving visualization, spatial accuracy, and surgical outcomes. MxR navigation (MRN) provides a cost-effective and intuitive alternative to conventional navigation systems, enabling stable spatial mapping and independent navigation. Despite this potential, implementing MRN is complex, creating a need for structured workflows and rigorous accuracy assessments. To address this gap, this protocol presents a structured, modular, and reproducible lab protocol based on the open-source platform 3D Slicer, enabling effective implementation and evaluation of developed MRN systems in neurosurgical contexts. This protocol clearly defines sequential steps, required inputs, and expected outputs for both visualization-oriented and navigation-oriented workflows. It encompasses preprocessing (anonymization, quality checks, multimodal image fusion), semi-automatic segmentation of anatomical structures (e.g., lesions, vessels, fiber tracts), and the generation of precise 3D surface models (STL or OBJ formats). For navigation scenarios, the protocol includes parameterization of fiducial landmarks, anatomical surfaces, and laser projections for precise virtual-to-physical registration. Accuracy and performance validation are rigorously assessed using virtual and physical static digital twins, generating quantitative displacement analyses and intuitive visualizations directly interpretable in clinical and research environments. The modular design and exclusive reliance on open-source software ensure reproducibility, flexibility, and broad interdisciplinary accessibility, benefiting users from diverse backgrounds with basic knowledge in anatomy, neurosurgery, and computer graphics.

## Introduction

Standard surgical navigation systems have fundamentally transformed neurosurgical practice by enabling precise localization of surgical instruments relative to patient anatomy [1,2]. However, conventional navigation prototypes typically rely on external

**Data availability statement:** The data underlying the results presented in this study have been previously published and made openly available in a public repository under the Creative Commons Attribution 4.0 International License (CC-BY 4.0). The complete dataset can be accessed via the following DOI: https://doi.org/10.6084/m9.figshare.24550732.v6.

**Funding:** Open Access funding provided by the Open Access Publishing Fund of Philipps-Universität Marburg.

**Competing interests:** The authors have declared that no competing interests exist.

monitors, forcing surgeons to frequently shift attention between the patient and the display, potentially compromising workflow, ergonomics, and accuracy [3–6].

Mixed Reality (MxR), first introduced by Milgram et al. [7], encompasses a continuum between fully physical and fully virtual environments, integrating virtual content into real-world settings (Augmented Reality, AR) or real-world elements into virtual environments (Augmented Virtuality, AV) [8]. In neurosurgery, MxR techniques—particularly AR-based technologies—have been integrated through surgical microscopes or head-mounted displays (HMDs) to integrate computer-generated images directly into the surgical view [1,9–15]. These integrated systems typically depend on external tracking devices, complex hardware setups, and extensive calibration, resulting in high costs and logistical challenges, particularly in resource-limited environments [1,11,16]. As an alternative, compact and cost-effective MxR hardware, such as projectors, smartphones, and tablets, has been explored [17–22]. However, these devices alone cannot reliably achieve stable spatial mapping and autonomous navigation.

In contrast, MxR navigation (MRN) leverages dedicated MxR-HMDs that possess inherent spatial mapping capabilities. This enables the autonomous, real-time alignment of virtual and physical data, effectively achieving navigation without external tracking [23–26]. These systems significantly simplify anatomical comprehension, improve the ergonomics, and enhance feasibility even in resource-constrained settings. Despite MRN's substantial potential, practical deployment and broader acceptance remain hindered by ongoing technical and practical challenges.

On the one hand, there is currently no unified workflow to efficiently and reliably convert traditional cross-sectional medical images into accurate three-dimensional (3D) virtual models for neurosurgical planning. Although advances in machine learning (ML) have improved automated segmentation, achieving precise, patient-specific 3D reconstructions remains challenging [27–29]. Numerous open-source platforms (e.g., 3D Slicer, ITK-SNAP, MITK) offer capabilities for image processing, segmentation, and navigational guidance [5,25,30–33]. However, crucial functionalities and training resources are distributed across different software packages, often forcing users to combine multiple tools to complete a single workflow. A recent comparative evaluation by Amla et al. examined several freely available platforms for computer-assisted interventions and found that while 3D Slicer and ITK-SNAP achieved the highest overall rankings [34], none provided a comprehensive solution for all tasks (see Table 1). This fragmentation of resources and capabilities complicates development and the learning curve and hampers reproducibility in advanced neurosurgical planning [6,26]. Accordingly, there is a clear need for an integrated platform that consolidates these tools into a single, streamlined workflow.

On the other hand, MRN systems, as emerging interdisciplinary technologies, require validation for technical feasibility and clinical reliability [23,25]. Existing studies significantly vary in development and validation approaches, reproducibility, and comparability, primarily due to the considerable challenges in assessing the actual spatial localization accuracy of virtual objects [35]. While some protocols claim to evaluate MRN system accuracy, differences in research contexts, objectives, and

**Table 1. Software rankings based on proposed criteria: 1 for best, 8 for worst. Reproduced from Amla et al. [34].**

| | 3D Slicer | Elastix | ITK-SNAP | MedInria | MeVisLab | MIPAV | MITK Workbench | Seg3D |
|---|---|---|---|---|---|---|---|---|
| Load times | 7 | NA | 1 | 5 | 6 | 3 | 4 | 2 |
| Stressload | 3 | NA | 1 | 4 | failed | failed | 2 | failed |
| Multi-tasking | 5 | 1 | 2 | 6 | 3 | 5 | 7 | NA |
| Range of functionalities | 1 | 8 | 6 | 5 | 3 | 2 | 4 | 7 |
| Extensibility | Yes | No | No | No | Yes | Yes | Yes | No |
| Effectiveness of the functionalities | | | | | | | | |
| - Image registration | 2 | 1 | 3 | 5 | 7 | 4 | 6 | 8 |
| - Image segmentation | 2 | NA | 4 | 7 | 6 | 5 | 1 | 3 |
| User-friendliness | 4 | 5 | 1 | 2 | 8 | 6 | 3 | 7 |
| Documentation | 1 | 4 | 3 | 5 | 6 | 8 | 2 | 7 |
| Technical support | 1 | 4 | 2 | 6 | 5 | 8 | 3 | 7 |

"failed" indicates software inability to load the large image "NA" denotes tests that could not be applied (Elastix lacked UI and cannot segment images; Seg3D's multi-tasking performance was not compared due to registration preprocessing requirements). Citation Note: Table 1 *is reproduced from Amla et al. (2024), published in Journal of Digital Imaging (37(1):386–401, DOI: 10.1007/s10278-023-00912-y), under the Creative Commons Attribution 4.0 International License (CC BY 4.0). The table's content is used here without any modifications, under the CC BY 4.0 terms.*

methodologies severely limit comparability [6,26,36–38]. Additionally, practical implementation often involves feasibility concerns; recruiting clinical cases significantly slows research progress, while commercially available anatomical head phantoms incur additional costs, especially disadvantageous for resource-constrained settings [39]. Furthermore, validating and analyzing MRN systems typically necessitates specialized computer-aided design (CAD) tools and custom-written code, imposing substantial skill requirements on users and further complicating the validation process.

To address these challenges, this protocol, informed by multiple peer-reviewed studies [23,25,39], proposes a structured, modular protocol for the open-source platform 3D Slicer, specifically designed for MRN implementation and evaluation in neurosurgical contexts. Existing 3D Slicer tutorials and community resources are largely module-centric and typically cover single tasks (e.g., segmentation or visualization), offering limited guidance on how to connect these components into an MRN-specific pipeline that supports registration, twin-based validation, and quantitative benchmarking. Importantly, this protocol couples this workflow with step-level time estimates and standardized benchmark results or visualizations, enabling direct replication and quantitative comparison across MRN implementations. Rather than introducing yet another isolated example workflow, the protocol: (i) unifies image import, multimodal preprocessing, anatomical and functional modelling, registration preparation, and accuracy evaluation into a single, end-to-end MRN workflow; (ii) embeds a static digital-twin validation paradigm that enables image-derived, non-contact, and repeatable accuracy measurements without external CAD software or custom scripts; and (iii) formalizes an accuracy framework that distinguishes interpolation from extrapolation behaviour and separates virtual-to-physical from virtual-to-perceptual error domains. Together, these elements provide a reproducible methodological standard for MRN development that can be applied to surgical planning, education, and system validation.

## Materials and methods

This section first provides a brief overview, followed by software and hardware requirements and key ethical considerations. Next, it describes the protocol's two main workflow categories: visualization and navigation. Finally, supporting materials (a step-by-step protocol, demonstration video, and troubleshooting notes) are provided to enhance its reproducibility.

## Protocol overview

This protocol encompasses sequential and modular steps, including data import, preprocessing, anatomical segmentation, visualization, patient registration for navigation, validation, and evaluation (see Fig 1). Data initially imported in Digital Imaging and Communications in Medicine (DICOM) format undergo anonymization, quality checks, and multimodal image fusion to ensure usability and reliability. Postprocessing involves anatomical segmentation of relevant structures such as lesions, vessels, and fiber tracts, as well as elements critical to surgical planning (e.g., puncture trajectories). The resulting segmented models are exported as standardized 3D virtual models in common surface file formats (e.g., STL and OBJ), which are compatible with virtual reality (VR), MxR, and 3D printing applications.

For navigation-oriented scenarios, specific anatomical landmarks and surface models are extracted to accurately align virtual environments with the physical surgical setting, enabling an image-to-patient registration. Additionally, static digital twin models — virtual counterparts paired with corresponding physical 3D-printed models — are created to facilitate

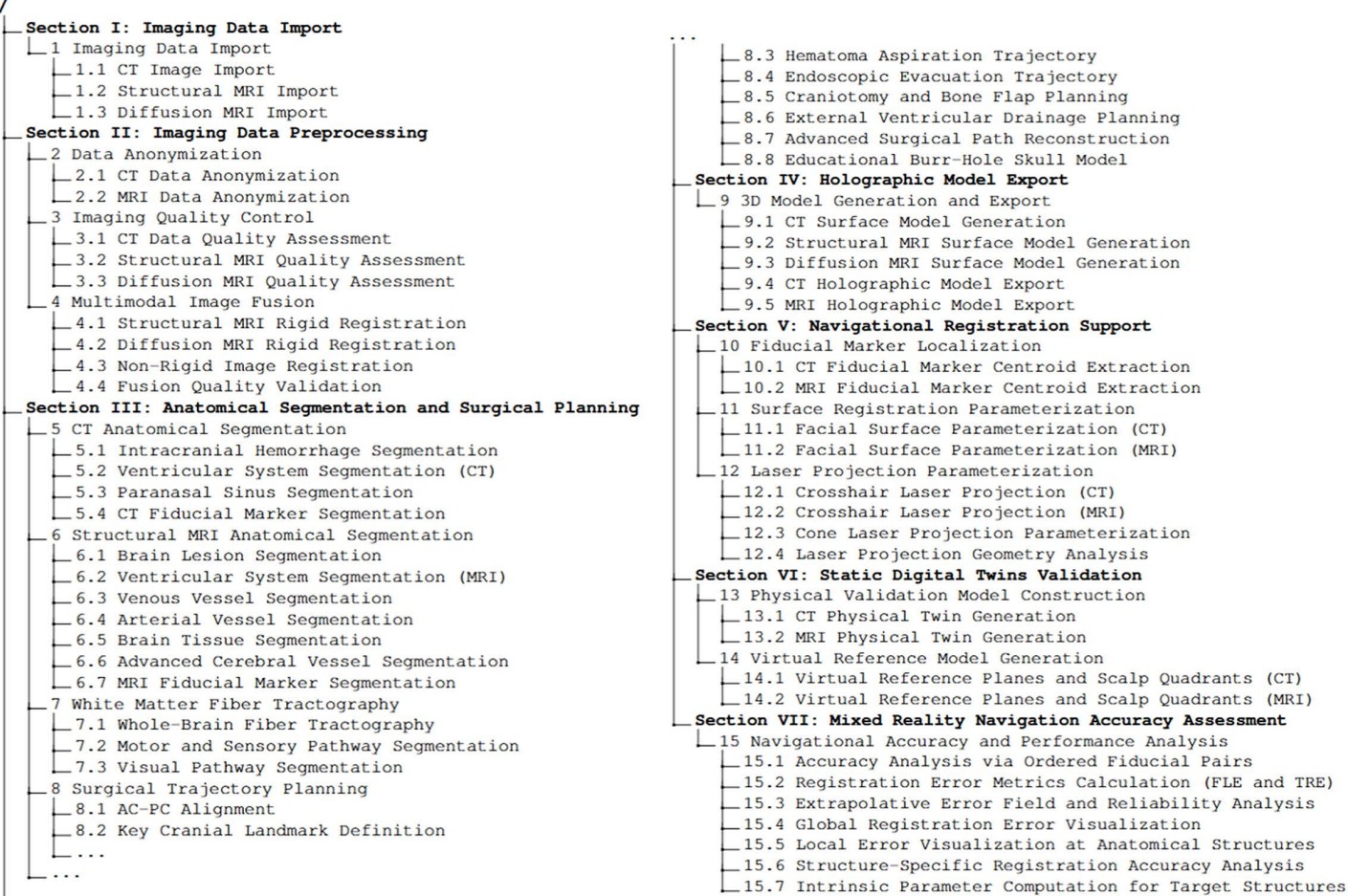

**Fig 1. Structure of the protocol.** A hierarchical overview illustrating the sequential arrangement of all major technical steps and procedures involved, clearly organized into different sections, including data import, preprocessing, segmentation, model generation, registration, and validation. Abbreviations: AC-PC = anterior commissure-posterior commissure; CT = computed tomography; FLE = fiducial localization error; MRI = magnetic resonance imaging; TRE = target registration error.

rigorous evaluation of system performance. Quantitative accuracy assessments, using integrated tools in 3D Slicer, ensure clear and intuitive verification of MRN system reliability.

To accommodate diverse user requirements, this protocol is organized into six modular pipelines tailored for different scenarios (detailed in the "Expected Results" section).

### Software and hardware

This protocol is predominantly based on the open-source medical image analysis software 3D Slicer (https://www.slicer.org, version 5.8.1), requiring no additional programming skills or custom scripting. However, output data, such as point coordinates and calculated metrics, can be exported in CSV format for further analysis in external software, including CloudCompare (https://www.cloudcompare.org/main.html) and MATLAB (https://www.mathworks.com/, The MathWorks Inc., Natick, Massachusetts, United States), ensuring extensibility.

System requirements (adapted from official 3D Slicer documentation, https://slicer.readthedocs.io/en/latest/user_guide/getting_started.html): The protocol runs effectively on modern Windows, macOS, or Linux systems (within the past five years), including virtual machines and Docker containers. Recommended hardware includes at least 8 GB of RAM, a dedicated GPU with OpenGL 3.2 support and sufficient VRAM to exceed the dataset size, and a display resolution of 1920 x 1080 pixels or higher. Input devices such as a standard three-button scroll mouse are recommended. An internet connection is also recommended for accessing software extensions, online documentation, datasets, and tutorials.

In this study, the practical demonstration cases were executed on a computer equipped with an Intel Core i7-1065G7 CPU at 1.30 GHz, 32 GB of RAM, integrated Intel Iris Plus Graphics, and running Windows 11 Pro (version 24H2).

### Ethical consideration

The demonstration data used in this study come from the publicly available MRN dataset on the Figshare platform [39], licensed under Creative Commons Attribution 4.0 International (CC BY 4.0, https://creativecommons.org/licenses/by/4.0/), and were accessed for this study on April 10, 2025. The MRN dataset was previously anonymized, and did not contain any personally identifiable patient information. As this research involved only the use of publicly accessible data without identifiable patient information, additional ethics approval and informed consent were not required. Because the study involved no fieldwork, patient recruitment, or access to clinical or experimental sites, no field site access permissions or collection permits were required.

### Visualization-oriented workflow

**Data import and preprocessing (Step 1 to Step 4).** *Step 1* imports the imaging datasets using the DICOM import function, which supports computed tomography (CT), structural magnetic resonance imaging (MRI), and diffusion-weighted MRI (dMRI). Using DICOM ensures the full fidelity of the image data. Users are strongly advised to avoid importing common image formats, such as PNG or JPEG (e.g., scanned film images), due to their limited grayscale depth, which can result in information loss and increased vulnerability to data corruption. Notably, functional MRI (fMRI) and ultrasound (US) data are not yet supported in this workflow. If necessary, *Step 2* anonymizes patient metadata via the "*Data*" module, replacing personal identifiers with pseudonyms. *Step 3* provides optional quality control (QC), e.g., by measuring the image signal-to-noise ratio (SNR) using the "*SNR Measurement*" extension and applying denoising if necessary to ensure sufficient image quality. In *Step 4*, all image volumes are co-registered (fused) into a patient-specific common coordinate system defined by the highest-resolution imaging dataset (usually a T1-weighted image) using the "*SlicerElastix*" module for a multimodal fusion [40,41]. To preserve data integrity, it is recommended to use the resulting transformation matrix to align images and segmentations (via the "*Transforms*" module) rather than creating any hardened resampled image volumes.

**Data postprocessing (Step 5 to Step 8).** *Steps 5 and 6* implement semi-automatic segmentation of key intracranial structures in the CT and MRI datasets, respectively, using "*Segment Editor*". The specific segmentation approach is adjusted based on the image characteristics of each target structure (e.g., distinguishing lesions, ventricles, or vasculature). Still, it generally relies on basic interactive tools, such as intensity thresholding and simple morphological operations. These methods are chosen for their efficiency and typically allow a given structure to be segmented within a few minutes. More advanced techniques (e.g., applying specialized filtering for vessel enhancement using, e.g., "*RVXVesselnessFilters*" module [42]) are available for experienced users, if needed. Overall, the segmentation strategy prioritizes obtaining reliable and sufficiently accurate labels with minimal imaging data and user interaction, thereby optimizing efficiency while preserving data integrity.

*Step 7* applies dMRI tractography to reconstruct major white-matter fiber pathways. Using "*SlicerDMRI*" toolkit (e.g., performing a tensor model fitting step followed by whole-brain tractography with the UKF algorithm), a full brain tracto-gram is generated [43–45]. Based on this, critical fiber bundles adjacent to the lesion can be identified by isolating tracts with regions of interest (ROIs). Notably, a more targeted seeding approach from specific anatomical landmarks may also be used to rapidly obtain key tracts, albeit with some loss of comprehensiveness.

*Step 8* incorporates surgical planning landmarks and the design of the trajectory. First, the patient's head orientation can be standardized (e.g., aligning along the anterior commissure-posterior commissure (AC-PC) line via an "*ACPC Alignment*" tool in "*SlicerNeuro*" extension module). Using this normalized orientation, common cranial entry points can be identified by adjusting slice views according to standard neurosurgical landmarks. Specific surgical target points of interest are then determined, either manually or by computing the centroids of relevant structures using the "*Segment Statistics*" module. Finally, prospective surgical paths are defined: these can be sketched as temporary segmentation labels or cre-ated as virtual model pathways (e.g., using the "*Port Placement*" tool within the "*SlicerIGT*" extension module) to represent planned instrument trajectories. Notably, *Step 8* specifically requires cranial landmarks derived from CT segmentation outputs (*Step 5*). Thus, Step 8 does not receive direct input from MRI segmentation (*Step 6*).

**Visualization (Step 9).** *Step 9* converts the finalized segmentation results into 3D surface models for visualization purposes. Using the "*Segmentations*" module, the binary label maps are converted to closed-surface meshes, yielding smooth 3D representations suitable for interactive rendering and surgical planning. To maintain geometric accuracy, any necessary spatial transformations should be applied to the label maps before conversion rather than to the surface meshes, to prevent topology distortions. The generated models' appearance can be adjusted as needed (e.g., setting colors or transparency via the "*Models*" module) and then visualized directly within the 3D viewer, which also supports stereoscopic 3D viewing, from simple red-cyan anaglyphs to fully immersive VR via the "*SlicerVirtualReality*" extension [46]. For MxR software on HMDs, the models can be exported in standard 3D file formats (e.g., OBJ or STL) and imported into external visualization environments (e.g., Unity3D for use with a HoloLens device).

## Navigation-oriented workflow

**Navigational registration (Step 10 to Step 12).** Steps 10–12 implement the registration components needed for surgical navigation. Three complementary registration strategies are prepared: (i) Fiducial-based registration (*Step 10*) by computing the 3D coordinates of each identifiable fiducial marker's centroid (using the "*Segment Statistics*" module on segmented reference landmarks) to serve as reference ground truth points; (ii) Surface-based registration (*Step 11*) by extracting the patient's skin surface as a binary label map and converting it to a 3D point cloud (e.g., in PLY format) that can be aligned to the patient's physical surface; and (iii) Laser projection-based registration (*Step 12*) by digitally replicating the geometry of surgical alignment laser patterns (such as crosshairs or conic sections) using the "*Markups*" and the "*Curve Maker*" module to facilitate laser-guided alignment [23,25]. Depending on the surgical setup and MxR software, one or more of these paradigms can be employed to synchronize the virtual and physical coordinate systems in real-time.

**Static digital twins validation (Step 13 and Step 14).** Steps 13 and 14 establish a pair of patient-specific "static digital twins" to calibrate and validate the navigation system. In ***Step 13***, a physical twin is created by segmenting the patient's imaging data (CT or MRI) to derive a 3D head model, incorporating any fiducial or projection markers, and then fabricating this model via 3D printing. In ***Step 14***, a corresponding virtual twin is generated: a simplified virtual representation of the same anatomy (for instance, featuring virtual fiducials or partitioned scalp regions to aid evaluation), which can be refined using the "*Dynamic Modeler*" module and exported for use in an MxR environment. These matched physical and virtual models serve as controlled corresponding pairs, allowing for a direct comparison of spatial positions and providing an independent and reusable basis for measuring the system's navigational accuracy.

**Accuracy assessment and presentation (Step 15).** ***Step 15*** provides a quantitative assessment of the MRN system's spatial accuracy using the twin models as ground truth references. After registering the MRN system to the static twins, the system's reported spatial coordinates are compared with the known reference positions on the virtual twin. Accuracy is assessed at two levels: (i) interpolation accuracy at the reference fiducial locations, and (ii) extrapolation accuracy across the broader workspace, reflecting how registration performance generalizes beyond the measured landmarks. These evaluations use the paired point sets and quantify discrepancies between virtual and physical coordinates. Together, they form a unified accuracy framework that distinguishes virtual-to-physical correspondence from virtual-to-perceptual discrepancies and can be applied across different MRN registration paradigms. Full metric definitions, rigid-transformation details, and extended visualization examples are provided in S4 Appendix.

## Step-by-step protocol

The protocol described in this peer-reviewed article is published on protocols.io (https://dx.doi.org/10.17504/protocols.io.kqdg327p1v25/v1) and is included for printing purposes as S1 File.

## Video demonstration

A detailed video demonstration of the protocol, using publicly accessible data from the MRN dataset, is available online at https://doi.org/10.6084/m9.figshare.29046527.v9, and is included as S2 Appendix.

## Troubleshooting

To assist users in quickly addressing potential operational difficulties, a concise troubleshooting guide is provided in S3 Table summarizing common issues, their likely causes, and recommended solutions.

## Expected results

This section summarizes the practical outputs generated by the protocol. It first introduces the six modular pipelines and their typical execution times, illustrating how different user groups can selectively apply the workflow. Representative use cases then demonstrate the protocol's application in multimodal neurosurgical planning, neurosurgical education, and MRN system validation. Standardized benchmark results derived from static digital twins are provided to support reproducibility and to establish reference accuracy levels for future MRN studies. Finally, the section positions the protocol within the broader landscape of existing tools and workflows and outlines its methodological contributions, limitations, and avenues for future development.

### General overview

Six specific-purpose protocol pipelines are developed, i.e., *Basic prototype*, *Advanced multimodal*, *Educational*, *Surgical navigation*, *Minimal developer*, and *Comprehensive research*, as detailed in Fig 2. The modular design enables the selective execution of steps and the installation of relevant software modules, tailored to meet diverse user needs. For various

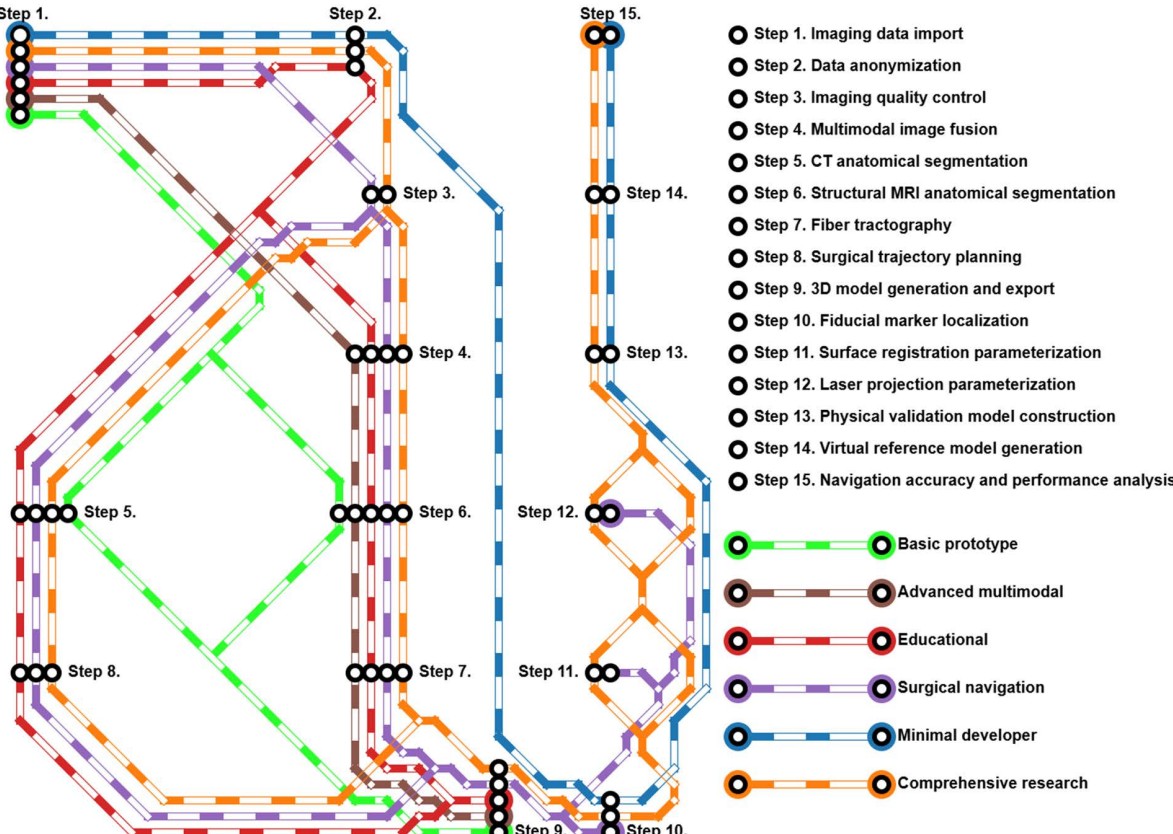

**Fig 2. Protocol pipelines and workflow integration.** Schematic representation illustrating the combinations of processing steps into distinct protocol pipelines tailored for various application scenarios, including *Basic prototype*, *Advanced multimodal*, *Educational*, *Surgical navigation*, *Minimal developer*, and *Comprehensive research* pipelines. Each color-coded pipeline illustrates the flexible and modular application of steps tailored to specific clinical, research, or educational purposes.

use cases, not all steps are mandatory. Table 2 outlines which steps apply to each pipeline. For instance, clinicians may primarily require visualization outputs for surgical planning (see Fig 3), while engineers or researchers can explore deeper spatial mappings and quantitative accuracy assessments. Notably, sub-steps within each pipeline are mainly independent and can be executed flexibly, allowing the simultaneous generation of multiple outputs. To complement the pipeline-level execution times in Table 2, Table 3 summarizes the typical duration of each major step. These values are derived from recorded demonstration videos and therefore represent upper-bound estimates; actual execution times in routine use may be shorter.

### Use-case: Multimodal neurosurgical planning

This illustrative case is based on patient data previously published in an open-access dataset under the CC-BY 4.0 license (Case No. 10, credit: Qi Z et al. (2024) Head model dataset for mixed reality navigation in neurosurgical interventions for intracranial lesions. *Sci Data* 11: 538. doi: 10.1038/s41597-024-03385-y). A 57-year-old female patient presented with progressive left lower-limb weakness. A contrast-enhanced T1-weighted image (T1-CE) identified a suspected metastatic ring-enhancing lesion in the right central region, which suggested a neurosurgical resection.

**Table 2. Overview of the MRN protocol pipelines.**

| Pipeline | Anonymization | Registration | Complexity | Steps involved† | Time [min/case] |
|---|---|---|---|---|---|
| Visualization-oriented | | | | | |
| – Basic prototype | No | No | 1/5 | 1, (5 or 6), 9 | Approx. 10 |
| – Advanced multimodal | No | No | 2/5 | 1, 4, 6, 7, 9 | Approx. 70 |
| – Educational | Yes | Optional | 3/5 | 1, 2, (5, 8 or 4, 6, 7), 9 | 60–75 |
| Navigation-oriented | | | | | |
| – Surgical navigation | No | Yes | 4/5 | 1, 3, (5, 8 or 4, 6, 7), 9, (10 or 11 or 12) | 60–75 |
| – Minimal developer | Yes | Yes | 4/5 | 1, 2, 10, 13–15 | 25–30 |
| – Comprehensive research | Yes | Yes | 5/5 | 1–3, (5, 8 or 4, 6, 7), 9, 10, (11 or 12),13–15 | 150–180 |

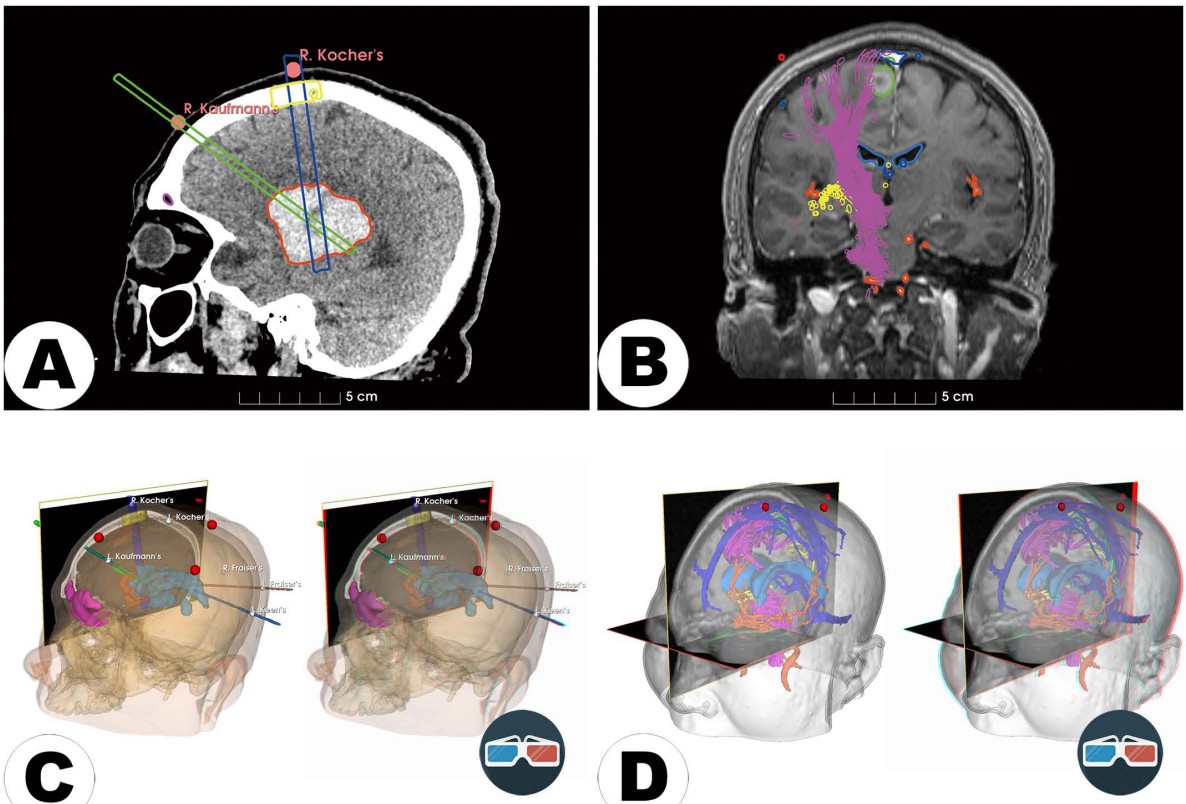

**Fig 3. Representative outputs of segmentation and visualization from pipelines.** A: Segmented anatomical structures from CT images. B: Segmented anatomical structures from MRI. C: Virtual reality (VR) and stereoscopic visualization of the segmented CT model (left: standard VR; right: 3D anaglyph viewable through red-cyan glasses). D: VR and stereoscopic visualization of the segmented MRI model (left: standard VR; right: 3D anaglyph).

Preoperative imaging data (T1-weighted image (T1WI), T1-CE, and diffusion tensor imaging (DTI)) were imported into the *Advanced multimodal* pipeline (***Step 1***). All images were aligned to the reference T1WI to ensure anatomical coherence (***Step 4***). Key anatomical structures, including the tumor, ventricles, cortical surface, and venous vessels, were segmented based on the T1-CE using semi-automatic segmentation tools ("*Segment Editor*" module, ***Step 6***), producing binary label maps. DTI data were then used to reconstruct whole-brain fiber tracts (fiber tractography, ***Step 7***),

**Table 3. Typical execution time for each major step of the protocol.**

| Step | Typical time [min] |
|---|---|
| Step 1: Imaging data import | 2–4 |
| Step 2: Data anonymization | <1 |
| Step 3: Image quality control | 1–3 |
| Step 4: Multimodal image fusion | 1–8 |
| Step 5: CT anatomical segmentation | 2–13 |
| Step 6: Structural MRI anatomical segmentation | 4–30 |
| Step 7: Fiber tractography | 22–25 |
| Step 8: Surgical trajectory planning | 1–43[†] |
| Step 9: 3D model generation and export | 5–8 |
| Step 10: Fiducial marker localization | 3 |
| Step 11: Surface registration parameterization | 8 |
| Step 12: Laser projection parameterization | 8–16 |
| Step 13: Physical validation model construction | 6–8 |
| Step 14: Virtual reference model generation | 4–5 |
| Step 15: Navigation accuracy and performance analysis | 32 |

[†]Wide ranges reflect case complexity and the number of targets/landmarks/entry points/trajectories. Values are derived from recorded demonstration videos and represent upper-bound estimates.

emphasizing critical pathways adjacent to the lesion, particularly the corticospinal tract (motor fibers, CST), somatosensory pathways (sensory fibers), and supplementary motor area (SMA) fibers. Segmentation outputs were subsequently transformed into closed-surface 3D models (***Step 9***), enabling detailed visualization and surgical planning in VR or MxR settings.

The generated 3D visualizations indicated that the lesion was situated beneath the postcentral gyrus (primary sensory cortex), causing notable cortical swelling and shifting nearby critical structures (see Fig 4). Specifically, both the central sulcus and the Rolandic vein were anteriorly displaced. Additionally, one of the well-developed Trolard veins was observed

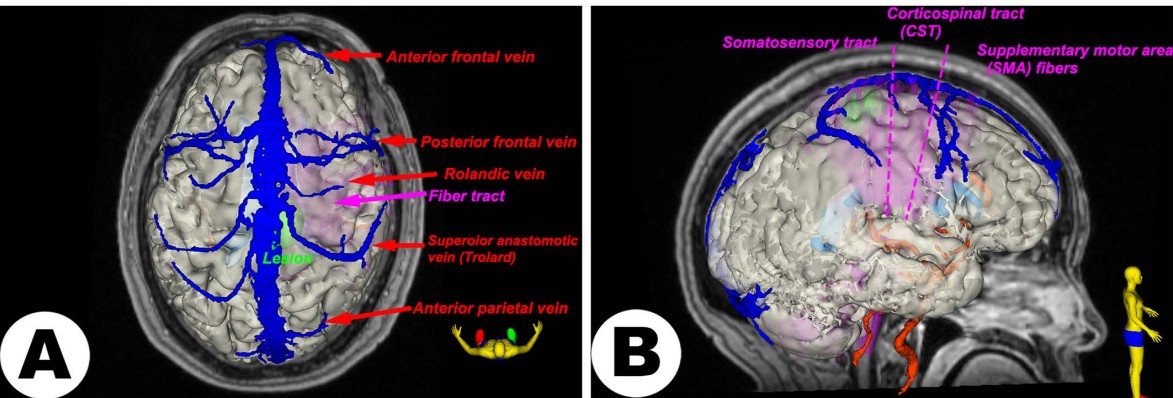

**Fig 4. Representative multimodal neurosurgical planning.** A preoperative multimodal 3D visualization highlighting critical anatomical structures for surgical planning of a right central metastatic brain lesion. The lesion (green) and adjacent functionally important structures—including superficial veins (blue), and fiber tracts (purple)—are depicted. A: Top view highlighting superficial veins and lesion proximity. B: Right-side view emphasizing fiber tract pathways, including the corticospinal tract (CST), somatosensory fibers, and supplementary motor area (SMA) fibers.

traversing anteriorly close to the planned surgical area. Based on these findings, an optimal surgical corridor was defined between the Trolard vein and the displaced somatosensory pathways, emphasizing preservation of venous drainage to minimize surgical complications. Given the presence of other prominent superficial anastomotic veins on the right hemisphere, intraoperative temporary clipping or fluorescence angiography could be performed to evaluate the potential sacrifice of this Trolard vein if necessary. This visualization-assisted surgical strategy aimed to maximize the probability of complete tumor resection while minimizing risks to critical venous drainage structures, functional cortical regions, and adjacent fiber tracts.

### Use-case: Neurosurgical education

This illustrative educational case is based on patient data previously published in an open-access dataset under the CC-BY 4.0 license (Case No. 17, credit: Qi Z et al. (2024) Head model dataset for mixed reality navigation in neurosurgical interventions for intracranial lesions. *Sci Data* 11: 538. doi: 10.1038/s41597-024-03385-y). A 63-year-old male presented with the acute onset of slurred speech and left-sided limb weakness. CT imaging revealed an acute intracerebral hemorrhage (ICH) in the right basal ganglia, serving as an illustrative case for common neurosurgical interventions such as hematoma aspiration (minimally invasive hematoma removal), endoscopic hematoma evacuation, external ventricular drainage (EVD, for cerebrospinal fluid drainage), or intracranial pressure (ICP) monitoring probe placement.

The preoperative CT data were imported into the *Educational* pipeline (**Step 1**). Key anatomical structures, including the hematoma, ventricular system, and frontal sinuses, were precisely segmented, generating binary label maps (**Step 5**). Subsequently, standardized anatomical alignment based on the AC-PC line was performed (**Step 8**), enabling accurate digital annotation of standard neurosurgical landmarks such as Kaufmann's, Kocher's, Keen's, and Frazier's points (common cranial entry points used in neurosurgical procedures [47]; see Fig 5A). Using these annotations, puncture sites and surgical skull models suitable for 3D printing were designed for practical training and educational purposes (see Fig 5B).

Binary label maps from the segmentation step were converted into closed-surface 3D models for enhanced visualization and immersive interaction in VR and MxR educational environments (**Step 9**). While precise spatial registration of virtual to physical models is optional in purely educational scenarios, performing registration when feasible, such as overlaying virtual models onto physical head phantoms or cadaveric specimens, is encouraged to provide an intuitive, hands-on experience, reinforcing trainees' understanding of neuroanatomical relationships and surgical strategies.

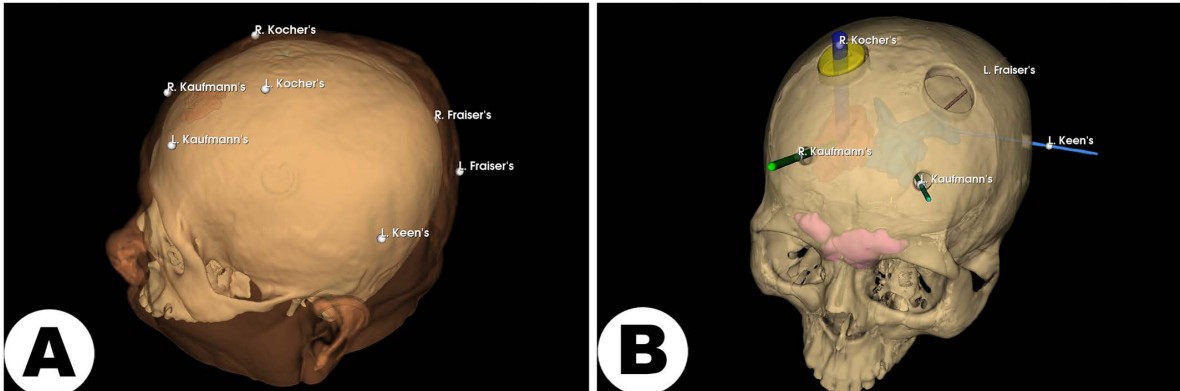

**Fig 5. Representative examples for neurosurgical education.** A: Digitally annotated standard cranial landmarks (Kaufmann, Kocher, Keen, Frazier points) on a realistic head phantom for educational use; B. An educational skull phantom demonstrating surgical entry points (burr holes) aligned with critical anatomical landmarks, suitable for 3D printing and hands-on training of common neurosurgical procedures.

### Use-case: MRN development, validation, and accuracy assessment

Registration in MRN fundamentally relies on common reference objects shared between virtual and physical spaces [39]. To enable automated or semi-automated registration, these reference objects must be parameterized and digitized into computationally tractable formats, thereby ensuring mathematically optimized, precise registration solutions. In this protocol, these reference objects are represented as coordinate point sets, including the centroid coordinates of fiducial markers (*Step 10*), the surface mesh vertices (*Step 11*), and the interpolated control points derived from laser-projection curves (*Step 12*). All reference object point sets are exported as ".fcsv" files, facilitating subsequent registration calculations.

Specifically, the MRN system calculates optimal spatial transformations by fitting known virtual point sets to corresponding physically acquired point sets. The *Surgical navigation* pipeline described in this protocol has been successfully utilized in multiple published and ongoing studies, demonstrating effective implementation of four distinct registration paradigms: fiducial-based registration [6,48], surface-based registration [49], crosshair-laser-based registration [23,25], and conical-laser-based registration (see Fig 6). These paradigms have been validated in at least 40 cases, including clinical patients and head phantoms [39].

The clinical reliability and effectiveness of MRN systems heavily depend on their spatial registration and localization accuracy. To rigorously validate MRN accuracy, this protocol employs a static digital twins approach, which involves paired physical and virtual anatomical reference models, providing a controlled, reproducible validation environment independent of actual patient data (*Comprehensive research* pipeline). Representative mixed-reality overlays from MRI- and CT-based cases are shown in Fig 7.

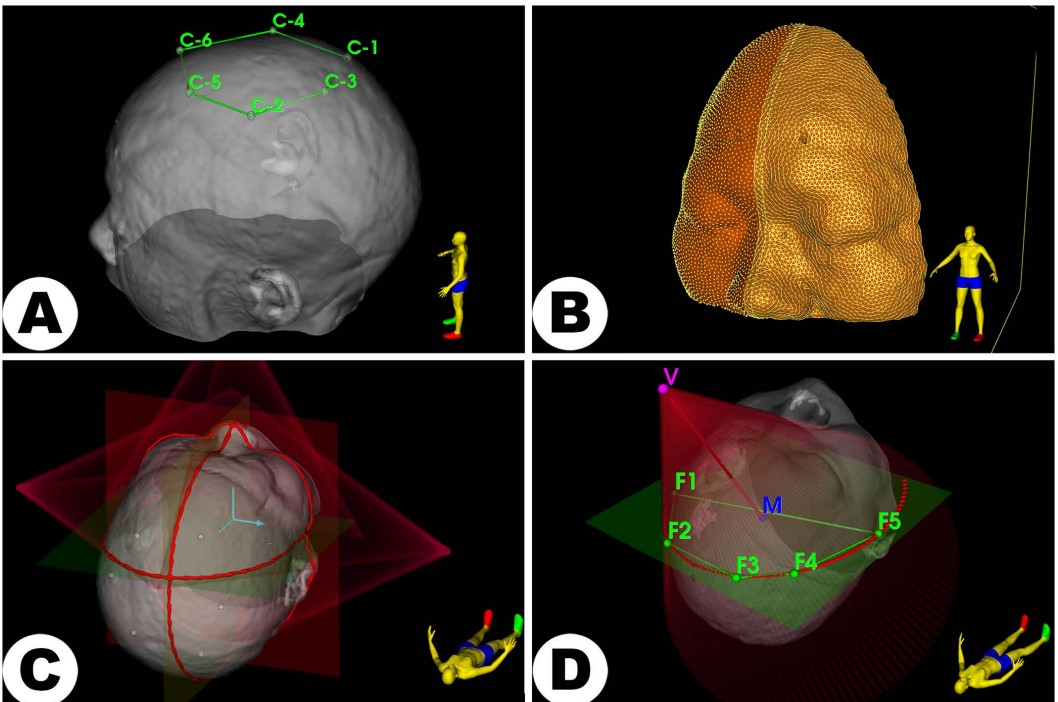

**Fig 6. Representative examples of reference object parameterization for MRN registration methods.** Navigation-oriented pipelines can be applied to develop registration techniques using: A: Physical artificial landmarks, B: Surface vertices and triangle meshes, C: Crosshair laser projections combined with anatomical planes, D: Conic laser projections and their projected non-degenerate conic sections (i.e., the curve produced when a plane intersects a cone at a non-trivial angle).

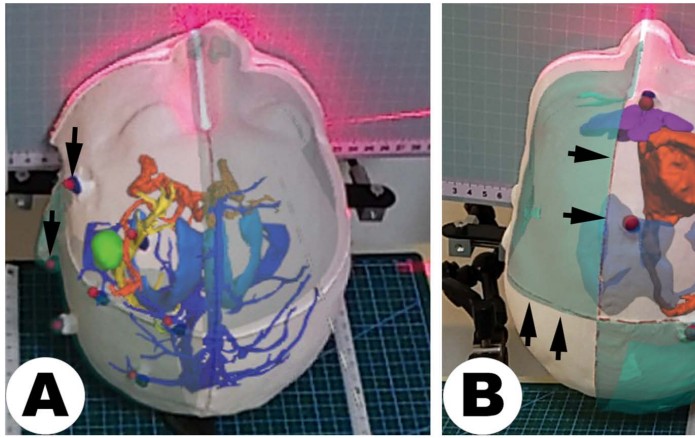

**Fig 7. Demonstration of virtual-to-physical alignment.** Photographs captured through the MRN system after registering virtual models to patient-specific 3D-printed head phantoms. Arrows highlight corresponding physical fiducials or lines and overlaid virtual anatomy, enabling qualitative verification of virtual-to-physical consistency before quantitative benchmarking. A: MRI-based multimodal overlay. B: CT-based overlay with scalp partition visualization.

**Step 15** uses paired point sets derived from the static twins to compute accuracy metrics that characterize MRN spatial performance; full metric definitions and extended examples are provided in S4 Appendix. To support reproducibility and cross-study comparability, we report protocol-linked reference benchmarks from our companion validation study of a laser crosshair simulator (LCS) registration method across 19 patient-specific head phantoms (21 lesions for *DSC* and $HD_{95}$) [25]. Summary statistics are provided in Table 4, distributions are visualized in Fig 8, and case-wise values are listed in Table 5, which together provide a practical baseline for sanity-checking and benchmarking future MRN implementations.

## Comparison with related tools and resources

In contrast to existing, fragmented resources and isolated tutorials, this protocol offers a coherent, well-defined workflow for MRN implementation and includes consolidated troubleshooting guidance for each step (see S3 Table).

Many open-source platforms support parts of the MRN workflow but differ in scope and usability. For example, ITK-SNAP is lightweight and effective for segmentation, yet it provides limited extensibility for building broader MRN pipelines

**Table 4. Quantitative benchmarking summary across all validation cases (n = 19).**

| Metric | M ± SD† |
|---|---|
| *FLE* [mm] | 1.9 ± 0.4 |
| *TRE* [mm] | 3.0 ± 0.5 |
| *FN* | 3.4 ± 1.7 |
| *FRE* [mm] | 2.1 ± 0.6 |
| *R* [°] | 1.5 ± 0.8 |
| *t* [mm] | 3.4 ± 1.7 |
| *DSC* | 0.83 ± 0.12 |
| $HD_{95}$ [mm] | 2.3 ± 1.0 |
| *fn* | 2.2 ± 0.9 |
| *fre* [mm] | 1.4 ± 0.7 |

†Mean ± Standard Deviation.

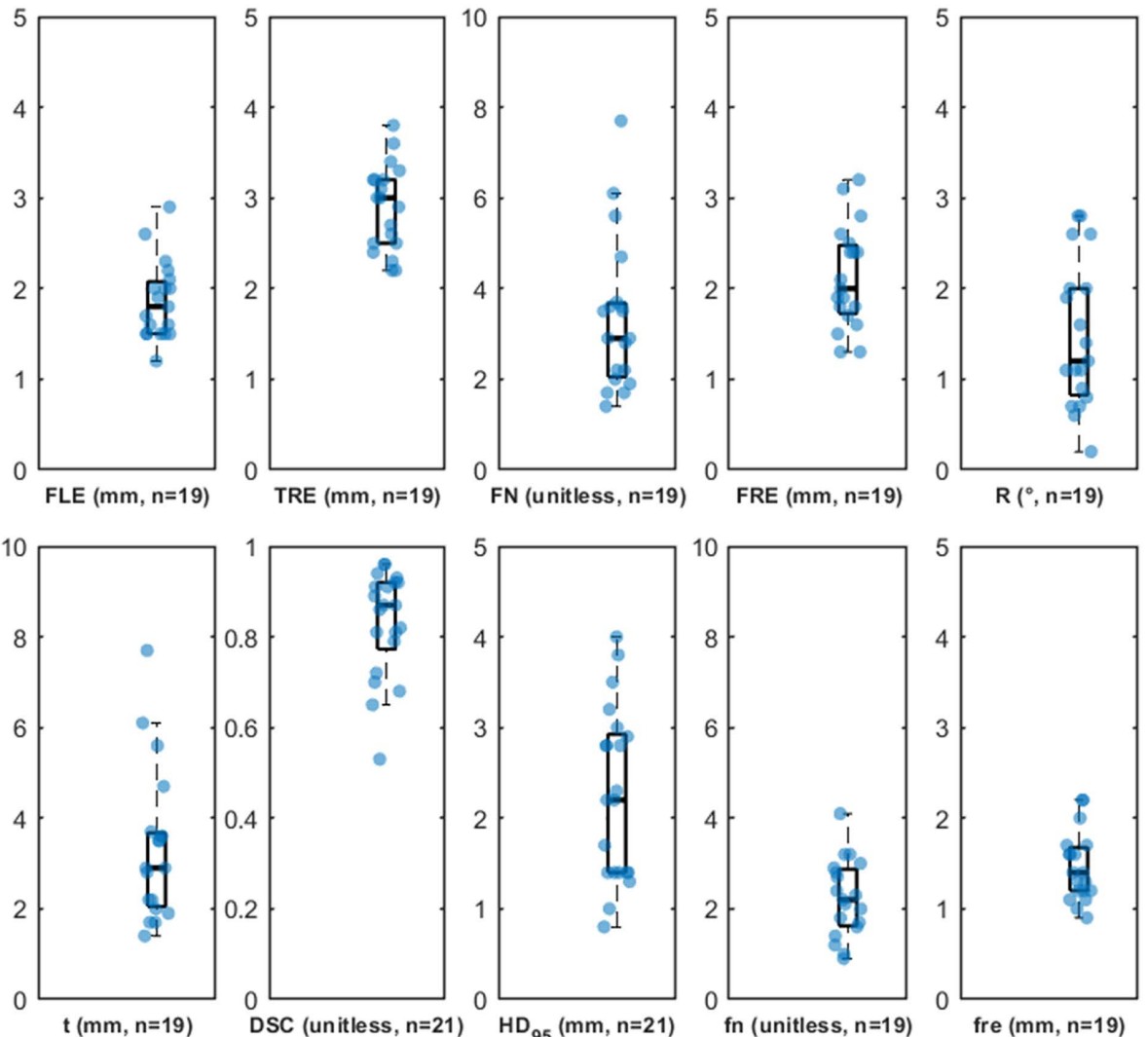

**Fig 8. Benchmark accuracy metrics across 19 cases and 21 lesions.** Boxplots with jittered scatter points summarize the distributions of *FLE*, *TRE*, *FN*, *FRE*, *R*, *t*, *DSC*, *HD₉₅*, *fn*, and *fre*. Each point corresponds to one validation case ($n = 19$) or, for lesion-specific structural agreement metrics (*DSC* and $HD_{95}$), one lesion ($n = 21$; two cases contained two lesions). Units and sample sizes are indicated in each panel. Boxes indicate the interquartile range (IQR) with the median as the central line; whiskers extend to the most extreme values within 1.5 IQR.

[30,31]. Other frameworks (e.g., MeVisLab) offer powerful functions but can impose steep learning curves for beginners (see Table 1) [34,50,51]. In MRN development, validation, and performance analysis, external CAD tools or custom scripts are often used, which increases technical demands and hampers reproducibility. 3D Slicer provides a practical compromise by combining core image computing functions with a large ecosystem of extensions, enabling segmentation, visualization, registration parameterization, and analysis within a single environment [52,53].

While 3D Slicer documentation and tutorials are extensive, they are typically feature- or module-centric and rarely describe how to assemble a complete MRN workflow end-to-end. The present protocol addresses this gap by defining six scenario-oriented pipelines with explicit step dependencies, inputs/outputs, and typical execution times, complemented by a printed step-by-step protocol (S1 File), video walkthroughs (S1 Appendix), and troubleshooting guidance (S3 Table).

**Table 5. Detailed case-wise quantitative benchmarking results.**

| Case | FLE [mm] | TRE [mm] | FN | FRE [mm] | R [°] | t [mm] | DSC† | HD95 [mm]† | fn | fre [mm] |
|------|----------|----------|-----|----------|-------|--------|------|------------|-----|----------|
| 01 | 2.3 | 3.3 | 2.9 | 3.1 | 2.0 | 2.9 | 0.81 | 2.8 | 2.2 | 1.6 |
| 02 | 2.2 | 2.7 | 3.5 | 2.4 | 0.7 | 3.5 | 0.81 | 2.9 | 1.7 | 2.2 |
| 03 | 1.7 | 2.4 | 2.0 | 1.8 | 0.9 | 2.0 | 0.91 | 1.4 | 1.0 | 1.3 |
| 04 | 1.6 | 2.5 | 6.1 | 1.8 | 2.8 | 6.1 | 0.82 | 3.8 | 2.4 | 1.4 |
| 05 | 1.5 | 2.9 | 2.2 | 2.1 | 0.6 | 2.2 | 0.89 | 2.2 | 2.3 | 1.3 |
| 06 | 2.6 | 2.6 | 2.8 | 1.7 | 1.2 | 2.8 | 0.96 | 1.4 | 3.2 | 2.0 |
| 07 | 1.6 | 3.6 | 2.9 | 2.4 | 1.1 | 2.9 | 0.91 | 2.8 | 1.8 | 1.4 |
| 08 | 1.9 | 3.8 | 3.7 | 3.2 | 1.6 | 3.7 | 0.68 [0.53] | 3.5 [4.0] | 2.1 | 1.1 |
| 09 | 2.1 | 3.2 | 5.6 | 2.8 | 2.6 | 5.6 | 0.65 | 3.0 | 2.8 | 1.6 |
| 10 | 1.5 | 3.4 | 7.7 | 2.5 | 2.6 | 7.7 | 0.79 | 2.2 | 2.7 | 1.2 |
| 11 | 1.5 | 3.0 | 3.5 | 1.9 | 2.0 | 3.5 | 0.87 | 1.7 | 3.0 | 1.1 |
| 12 | 2.0 | 2.3 | 1.7 | 1.5 | 1.4 | 1.7 | 0.93 | 1.0 | 2.0 | 1.6 |
| 13 | 2.9 | 3.2 | 3.6 | 2.6 | 1.1 | 3.6 | 0.70 | 2.8 | 3.2 | 2.2 |
| 14 | 1.2 | 3.0 | 3.6 | 2.4 | 2.8 | 3.6 | 0.87 [0.72] | 1.4 [2.3] | 1.4 | 0.9 |
| 15 | 2.0 | 2.5 | 4.7 | 2.0 | 1.9 | 4.7 | 0.86 | 3.2 | 4.1 | 1.7 |
| 16 | 1.5 | 3.2 | 1.7 | 1.6 | 1.1 | 1.7 | 0.92 | 1.4 | 0.9 | 1.2 |
| 17 | 2.0 | 2.2 | 1.4 | 1.3 | 0.7 | 1.4 | 0.96 | 0.8 | 1.6 | 1.7 |
| 18 | 1.8 | 2.2 | 2.2 | 1.3 | 0.8 | 2.2 | 0.92 | 1.4 | 2.9 | 1.2 |
| 19 | 1.5 | 3.1 | 1.9 | 1.9 | 0.2 | 1.9 | 0.94 | 1.3 | 1.2 | 1.0 |

†In cases 08 and 14, which each contain two lesions, the metrics outside brackets denote the larger lesion, and those in brackets denote the smaller lesion.

## Positioning within the literature

Within the broader context of neurosurgical 3D visualization and navigation, existing studies can be broadly categorized into three domains: preoperative surgical planning [54–57], educational or rehearsal-based visualization [58,59], and intraoperative navigation systems [60,61], as well as hybrid approaches combining these purposes [62,63]. The present protocol is designed to be compatible with all of these applications, providing an end-to-end methodological framework that can be directly applied to surgical planning, training, and navigation validation.

For visualization and educational applications, most reported workflows follow a similar data-processing chain that includes sequence alignment, segmentation, 3D surface reconstruction, and model optimization to enhance realism. These processes commonly involve transforming multimodal medical image coordinates into an anatomically aligned space, segmenting relevant structures, converting label maps into geometric surface models, and applying smoothing or texturing to improve visual quality [58]. Some studies further enrich the models with advanced attributes to simulate tissue deformation [58], or introduce parametric nodes to quantitatively assess trainees' performance during simulated procedures [57]. Despite these technical advances, the underlying procedures are far from straightforward: they require close collaboration between experts in neurosurgery, anatomy, radiology, computer graphics, and engineering [58]. Detailed step-by-step procedures are rarely disclosed, processing times and complexity are seldom reported, and open-source reproducibility is often lacking.

For navigation-oriented studies, validation frequently relies on paired virtual-to-physical reference models (digital twins), using cadaveric heads, patient-specific 3D-printed models, or the patient's own anatomy [58,60,61]. These approaches can be limited by the time and regulatory overhead of recruiting clinical cases [39,64], the cost of phantoms [39,64], and the invasiveness of probe-based measurements, which may reduce repeatability [5,59]. By contrast, the present protocol

employs static digital twins derived from publicly available datasets and supports non-contact measurements within MxR, enabling efficient repeated evaluations [23,25,39]. In parallel, it consolidates heterogeneous accuracy reporting into a unified taxonomy that separates interpolation from extrapolation and distinguishes virtual-to-physical from virtual-to-perceptual domains (technical details in S4 Appendix) [39].

## Limitations and future directions

Despite its strengths, this protocol has several limitations. Firstly, the protocol assumes some baseline knowledge of neuroanatomy, neurosurgical concepts, and basic 3D graphics, creating a relatively steep initial learning curve for beginners. Nevertheless, this challenge might be partially compensated by the step-by-step protocol with detailed instructions (see S1 File), supplementary video demonstrations for visual clarification (see S1 Appendix), and troubleshooting guidance to address common practical difficulties that the user may encounter (see S3 Table). Evaluating user learning curves and refining documentation based on feedback are ongoing efforts to mitigate this issue.

Secondly, the current protocol is limited to static digital twins, which serve as fixed geometric references and do not track real-time deformations [65]. This static design intentionally reflects the intended scope of the present work: to establish a reproducible, rigid baseline for evaluating MRN registration accuracy under controlled conditions. Such static twins remain indispensable, as any future dynamic or deformable MRN system must first demonstrate reliable performance in a stable and repeatable reference space before more complex, time-varying validations become meaningful. However, static twins cannot account for intraoperative tissue deformation. Accordingly, the present protocol is intended as a rigid baseline primarily for phantom-based validation and initial registration assessment (i.e., before substantial intraoperative deformation), rather than for modeling post-dural-opening brain shift or other time-varying tissue dynamics. At the cranial level, subtle skull deflection introduced by fixation pins or clamps may cause small but relevant rigid offsets before dural opening [1]. At deeper anatomical layers, dynamic intracranial deformation—including gravity-induced brain shift, cerebrospinal fluid egress, retraction effects, and resection-cavity collapse—cannot be represented by static replicas [1]. Progress toward dynamic digital twins will require coupling intraoperative measurements with computational deformation models. Several technological pathways are compatible with, and may extend, the protocol presented here. Real-time surface acquisition from intraoperative ultrasound or depth cameras can provide time-varying point clouds suitable for non-rigid registration using existing tools such as Elastix B-spline transforms [65]. Modules such as *Segment Mesher* and *SlicerBiomech* enable the creation of patient-specific tetrahedral meshes for finite-element solvers, whose predicted deformation fields can be reimported to warp preoperative volumes [66,67]. AI-based monocular depth-estimation approaches offer an additional low-cost means for reconstructing intraoperative cortical surfaces when advanced imaging modalities are unavailable [58,68]. Together, these avenues outline a coherent progression toward dynamic digital twins based on (i) real-time surface updates, (ii) biomechanical or data-driven deformation modeling, and (iii) continuous refinement of virtual-to-physical alignment. Importantly, the accuracy taxonomy and modular design of the present protocol are compatible with future extensions toward time-varying, 4D evaluations.

Thirdly, the protocol currently relies heavily on high-quality multimodal imaging data (e.g., high-resolution structural and diffusion MRI), which might be challenging to obtain in resource-limited settings. An ideal protocol should exhibit robustness by minimizing sensitivity to variations in data quality and quantity. Future improvements should therefore focus on adaptive preprocessing and robust algorithms that can handle suboptimal or incomplete datasets, thereby enhancing applicability across diverse clinical environments.

Fourthly, current accuracy assessments predominantly focus on initial spatial registration, often involving a subjective judgment component. Future iterations of the protocol should include systematic assessments of temporal stability, and leverage real-time 4D data streams to enhance accuracy. For example, the extension module "*Sequences*" in 3D Slicer can facilitate spatio-temporal calibration, replay, and analysis. This would provide new accuracy metrics and more intuitive visualizations of time-varying changes, ultimately enhancing the protocol's clinical applicability. Additionally, further

automation, such as machine learning algorithms, may eventually automate registration and quality assessment steps, significantly reducing manual workload and variability.

The future integration of this protocol via an interface with hospital information systems, including electronic health records, will be critical for broader clinical deployment. For instance, leveraging an interoperability standard such as Fast Healthcare Interoperability Resources (FHIR) would facilitate seamless data flow and cross-system interoperability. This forward-looking alignment with FHIR would enable the protocol to be integrated into routine clinical workflows, thereby overcoming the limitations of its current standalone implementation.

## Conclusion

The protocol presents a comprehensive MxR visualization and navigation protocol built upon the 3D Slicer platform, addressing critical gaps left by existing, fragmented tutorials and resources. It reduces entry barriers for users from diverse backgrounds and empowers them to effectively leverage MRN technologies, accelerating their clinical adoption and technological advancement in precision neurosurgery.

## Supporting information

**S1 File. Step-by-step protocol.** A step-by-step protocol, also available on protocols.io (https://dx.doi.org/10.17504/protocols.io.kqdg327p1v25/v1).
(PDF)

**S2 Appendix. Step-by-step video-based instructions.** The video-based instructions referenced by timestamps throughout the protocol, also available on figshare.com. https://doi.org/10.6084/m9.figshare.29046527.v9
(TXT)

**S3 Table. Troubleshooting Guide for the Step-by-step Protocol.** Listed are common problems encountered in each step, possible underlying causes, and recommended corrective actions to ensure smooth protocol execution.
(PDF)

**S4 Appendix. Technical details of the accuracy assessment framework.** Provided are detailed mathematical definitions of the accuracy metrics, displacement field computation, and additional visualization examples.
(PDF)

## Acknowledgments

We would like to express our sincere appreciation to Dr. Hui Zhang for her invaluable assistance during this research.

## Author contributions

**Conceptualization:** Ziyu Qi, Miriam H. A. Bopp.

**Data curation:** Ziyu Qi.

**Formal analysis:** Ziyu Qi.

**Funding acquisition:** Christopher Nimsky, Miriam H. A. Bopp.

**Investigation:** Ziyu Qi.

**Methodology:** Ziyu Qi.

**Project administration:** Ziyu Qi, Christopher Nimsky, Miriam H. A. Bopp.

**Resources:** Ziyu Qi, Christopher Nimsky, Miriam H. A. Bopp.

**Software:** Ziyu Qi.

**Supervision:** Christopher Nimsky, Miriam H. A. Bopp.

**Validation:** Christopher Nimsky, Miriam H. A. Bopp.

**Visualization:** Ziyu Qi.

**Writing – original draft:** Ziyu Qi.

**Writing – review & editing:** Christopher Nimsky, Miriam H. A. Bopp.

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
