## [Decision Letter · Decision Letter 0]

22 Oct 2025

Dear Dr. Qi,

Thank you for submitting your manuscript to PLOS ONE. After careful consideration, we feel that it has merit but does not fully meet PLOS ONE’s publication criteria as it currently stands. Therefore, we invite you to submit a revised version of the manuscript that addresses the points raised during the review process.

If applicable, we recommend that you deposit your laboratory protocols in protocols.io to enhance the reproducibility of your results. Protocols.io assigns your protocol its own identifier (DOI) so that it can be cited independently in the future. For instructions see: https://journals.plos.org/plosone/s/submission-guidelines#loc-laboratory-protocols. Additionally, PLOS ONE offers an option for publishing peer-reviewed Lab Protocol articles, which describe protocols hosted on protocols.io. Read more information on sharing protocols at. Additionally, PLOS ONE offers an option for publishing peer-reviewed Lab Protocol articles, which describe protocols hosted on protocols.io. Read more information on sharing protocols at https://plos.org/protocols?utm_medium=editorial-email&utm_source=authorletters&utm_campaign=protocols..

We look forward to receiving your revised manuscript.

Kind regards,

Muhammad Mohsin Khan

Academic Editor

PLOS ONE

Journal Requirements:

3. We noted in your submission details that a portion of your manuscript may have been presented or published elsewhere. “Table 1 and Figure 8 are reproduced under CC-BY 4.0 license, credit:

Table 1: Amla et al. (2024), published in Journal of Digital Imaging (37(1):386–401, DOI: 10.1007/s10278-023-00912-y)

Figure 8: Qi Z et al. (2024) Head model dataset for mixed reality navigation in neurosurgical interventions for intracranial lesions. Sci Data 11: 538. doi: https://doi.org/10.1038/s41597-024-03385-y ”. Please clarify whether this [conference proceeding or publication] was peer-reviewed and formally published. If this work was previously peer-reviewed and published, in the cover letter please provide the reason that this work does not constitute dual publication and should be included in the current manuscript.

4. We note you have not yet provided a protocols.io PDF version of your protocol and/or a protocols.io DOI. When you submit your revision, please provide a PDF version of your protocol as generated by protocols.io (the file will have the protocols.io logo in the upper right corner of the first page) as a Supporting Information file. The filename should be S1_file.pdf, and you should enter “S1 File” into the Description field. Any additional protocols should be numbered S2, S3, and so on. Please also follow the instructions for Supporting Information captions [https://journals.plos.org/plosone/s/supporting-information#loc-captions]. The title in the caption should read: “Step-by-step protocol, also available on protocols.io.”

Please assign your protocol a protocols.io DOI, if you have not already done so, and include the following line in the Materials and Methods section of your manuscript: “The protocol described in this peer-reviewed article is published on protocols.io (https://dx.doi.org/10.17504/protocols.io.[...]) and is included for printing purposes as S1 File.” You should also supply the DOI in the Protocols.io DOI field of the submission form when you submit your revision.

If you have not yet uploaded your protocol to protocols.io, you are invited to use the platform’s protocol entry service [https://www.protocols.io/we-enter-protocols] for doing so, at no charge. Through this service, the team at protocols.io will enter your protocol for you and format it in a way that takes advantage of the platform’s features. When submitting your protocol to the protocol entry service please include the customer code PLOS2022 in the Note field and indicate that your protocol is associated with a PLOS ONE Lab Protocol Submission. You should also include the title and manuscript number of your PLOS ONE submission.

Additional Editor Comments:

The manuscript presents a detailed, open-source protocol for mixed-reality (MR) visualization and navigation using 3D Slicer. As a Lab Protocol, its primary purpose is to deliver a reproducible, step-by-step workflow rather than to test a scientific hypothesis. The topic is clinically relevant, and the documentation is significant.

However, several issues limit the manuscript’s impact and should be addressed before acceptance.

1. Scope and Innovation within a Lab Protocol

Even allowing for the procedural focus of a Lab Protocol, the manuscript remains largely an integration of existing 3D Slicer modules (segmentation, surface modeling, fiducial-based registration). The unique methodological advance—beyond assembling known tools—is not sufficiently articulated.

Please clearly state what is genuinely novel (eg, a distinctive integration strategy, the static digital-twin validation framework, or standardized accuracy metrics) and explain why this consolidation itself represents a meaningful advance over existing Slicer tutorials or comparable platforms such as MITK or MeVisLab.

2. Quantitative Validation and Benchmarking

The protocol commendably includes a built-in accuracy-assessment framework (TRE, FLE, displacement fields, Dice coefficient, HD95) and describes static digital-twin validation in detail. Despite these definitions, the manuscript provides no aggregated multi-case accuracy statistics, inter-observer reproducibility data, or comparative benchmarking against commercial navigation systems or other open-source pipelines. The authors need to add at least limited multi-case results or explicitly reference companion studies that report quantitative performance using this protocol to substantiate reliability and reproducibility.

3. Static Digital Twin Limitation

The protocol relies exclusively on static digital twins for accuracy testing. Static models cannot capture intraoperative brain shift or dynamic anatomical changes—key challenges for neurosurgical navigation. Discuss the clinical implications of this limitation and outline potential extensions (eg, real-time sensor integration, dynamic or “shadow” digital twins).

4. Positioning Within the Literature

Similar end-to-end MR workflows using 3D Slicer or related platforms have been described. The manuscript does not adequately compare its efficiency, accuracy, or ease of adoption with these prior efforts. Provide a critical, evidence-based comparison to related protocols like those I recommended below, highlighting concrete advantages such as reduced processing time, improved accuracy, or lower hardware costs.

Recommended References

1. Hanalioglu S, Romo NG, Mignucci-Jiménez G, et al. Development and Validation of a Novel Methodological Pipeline to Integrate Neuroimaging and Photogrammetry for Immersive 3D Cadaveric Neurosurgical Simulation. Front Surg. 2022;9:878378. doi:10.3389/fsurg.2022.878378. PMID:35651686; PMCID:PMC9149243.

2. Qi Z, Jin H, Wang Q, et al. Head model dataset for mixed reality navigation in neurosurgical interventions for intracranial lesions. Sci Data. 2024;11:538. doi:10.1038/s41597-024-03385-y. PMID:38796526; PMCID:PMC11329879.

3. Colombo E, Regli L, Esposito G, et al. Mixed Reality for Cranial Neurosurgical Planning: A Single-Center Applicability Study With the First 107 Subsequent Holograms. Oper Neurosurg (Hagerstown). 2023;26(5):551-558. doi:10.1227/ons.0000000000001033. PMID:38156882; PMCID:PMC11008664.

4. Hanalioglu S, Gurses ME, Mignucci-Jiménez G, et al. Infragalenic triangle as a gateway to dorsal midbrain and posteromedial thalamic lesions: descriptive and quantitative analysis of microsurgical anatomy. J Neurosurg. 2023;140(3):866-879. doi:10.3171/2023.6.JNS222871. PMID:37878005.

5. Yang Z, Su X, Yuan Z, et al. Mixed reality holographic navigation for intracranial lesions using HoloLens 2: A pilot study and literature review. Clin Neurol Neurosurg. 2025;256:109030. doi:10.1016/j.clineuro.2025.109030. Epub 2025 Jun 24. PMID:40570755.

6. Isikay I, Cekic E, Baylarov B, Tunc O, Hanalioglu S. Narrative review of patient-specific 3D visualization and reality technologies in skull base neurosurgery: enhancements in surgical training, planning, and navigation. Front Surg. 2024;11:1427844. doi:10.3389/fsurg.2024.1427844. PMID:39081485; PMCID:PMC11287220.

7. González-López P, Kuptsov A, Gómez-Revuelta C, et al. The Integration of 3D Virtual Reality and 3D Printing Technology as Innovative Approaches to Preoperative Planning in Neuro-Oncology. J Pers Med. 2024;14(2):187. doi: 10.3390/jpm14020187. PMID:38392620; PMCID:PMC10890029.

8. Shi Z, Peng Y, Gao X, et al. Translating high-precision mixed reality navigation from lab to operating room: design and clinical evaluation. BMC Surg. 2025;25(1):331. doi:10.1186/s12893-025-03096-0. PMID:40751242; PMCID:PMC12315425.

9. Hanalioglu S, Gurses ME, Baylarov B, Tunc O, Isikay I, Cagiltay NE, Tatar I, Berker M. Quantitative assessment and objective improvement of the accuracy of neurosurgical planning through digital patient-specific 3D models. Front Surg. 2024;11:1386091. doi:10.3389/fsurg.2024.1386091. PMID:38721022; PMCID:PMC11076751.

10. Kantak PA, Bartlett S, Chaker A, et al. Augmented Reality Registration System for Visualization of Skull Landmarks. World Neurosurg. 2024;182:369. doi:10.1016/j.wneu.2023.11.110. PMID:38013107.

11. Qi Z, Bopp MHA, Nimsky C, et al. A Novel Registration Method for a Mixed Reality Navigation System Based on a Laser Crosshair Simulator: A Technical Note. Bioengineering (Basel). 2023;10(11):1290. doi:10.3390/bioengineering10111290. PMCID:PMC10602959.

12. Begagić E, Bečulić H, Pugonja R, et al. Augmented Reality Integration in Skull Base Neurosurgery: A Systematic Review. Medicina (Kaunas). 2024;60(2):335. doi:10.3390/medicina60020335. PMID:38399622; PMCID:PMC10889940.

Recommendation

Major Revision. The authors should (1) articulate the distinct methodological contribution expected of a Lab Protocol, (2) strengthen quantitative validation with multi-case or companion data, (3) address the static-twin limitation, and (4) situate their workflow more rigorously within current mixed-reality navigation literature.

Reviewers' comments:

Reviewer's Responses to Questions

**Comments to the Author**



Reviewer #1: Yes

2. Has the protocol been described in sufficient detail?

To answer this question, please click the link to protocols.io in the Materials and Methods section of the manuscript (if a link has been provided) or consult the step-by-step protocol in the Supporting Information files.

Reviewer #1: Partly

3. Does the protocol describe a validated method?

Reviewer #1: Yes

4. If the manuscript contains new data, have the authors made this data fully available?

Reviewer #1: N/A

**5. Is the article presented in an intelligible fashion and written in standard English?**

Reviewer #1: Yes

Reviewer #1: The manuscript presents a detailed, open-source protocol for mixed-reality (MR) visualization and navigation using 3D Slicer. As a Lab Protocol, its primary purpose is to deliver a reproducible, step-by-step workflow rather than to test a scientific hypothesis. The topic is clinically relevant, and the documentation is significant.

However, several issues limit the manuscript’s impact and should be addressed before acceptance.

1. Scope and Innovation within a Lab Protocol

Even allowing for the procedural focus of a Lab Protocol, the manuscript remains largely an integration of existing 3D Slicer modules (segmentation, surface modeling, fiducial-based registration). The unique methodological advance—beyond assembling known tools—is not sufficiently articulated.

Please clearly state what is genuinely novel (eg, a distinctive integration strategy, the static digital-twin validation framework, or standardized accuracy metrics) and explain why this consolidation itself represents a meaningful advance over existing Slicer tutorials or comparable platforms such as MITK or MeVisLab.

2. Quantitative Validation and Benchmarking

The protocol commendably includes a built-in accuracy-assessment framework (TRE, FLE, displacement fields, Dice coefficient, HD95) and describes static digital-twin validation in detail. Despite these definitions, the manuscript provides no aggregated multi-case accuracy statistics, inter-observer reproducibility data, or comparative benchmarking against commercial navigation systems or other open-source pipelines. The authors need to add at least limited multi-case results or explicitly reference companion studies that report quantitative performance using this protocol to substantiate reliability and reproducibility.

3. Static Digital Twin Limitation

The protocol relies exclusively on static digital twins for accuracy testing. Static models cannot capture intraoperative brain shift or dynamic anatomical changes—key challenges for neurosurgical navigation. Discuss the clinical implications of this limitation and outline potential extensions (eg, real-time sensor integration, dynamic or “shadow” digital twins).

4. Positioning Within the Literature

Similar end-to-end MR workflows using 3D Slicer or related platforms have been described. The manuscript does not adequately compare its efficiency, accuracy, or ease of adoption with these prior efforts. Provide a critical, evidence-based comparison to related protocols like those I recommended below, highlighting concrete advantages such as reduced processing time, improved accuracy, or lower hardware costs.

Recommended References

1. Hanalioglu S, Romo NG, Mignucci-Jiménez G, et al. Development and Validation of a Novel Methodological Pipeline to Integrate Neuroimaging and Photogrammetry for Immersive 3D Cadaveric Neurosurgical Simulation. Front Surg. 2022;9:878378. doi:10.3389/fsurg.2022.878378. PMID:35651686; PMCID:PMC9149243.

2. Qi Z, Jin H, Wang Q, et al. Head model dataset for mixed reality navigation in neurosurgical interventions for intracranial lesions. Sci Data. 2024;11:538. doi:10.1038/s41597-024-03385-y. PMID:38796526; PMCID:PMC11329879.

3. Colombo E, Regli L, Esposito G, et al. Mixed Reality for Cranial Neurosurgical Planning: A Single-Center Applicability Study With the First 107 Subsequent Holograms. Oper Neurosurg (Hagerstown). 2023;26(5):551-558. doi:10.1227/ons.0000000000001033. PMID:38156882; PMCID:PMC11008664.

4. Hanalioglu S, Gurses ME, Mignucci-Jiménez G, et al. Infragalenic triangle as a gateway to dorsal midbrain and posteromedial thalamic lesions: descriptive and quantitative analysis of microsurgical anatomy. J Neurosurg. 2023;140(3):866-879. doi:10.3171/2023.6.JNS222871. PMID:37878005.

5. Yang Z, Su X, Yuan Z, et al. Mixed reality holographic navigation for intracranial lesions using HoloLens 2: A pilot study and literature review. Clin Neurol Neurosurg. 2025;256:109030. doi:10.1016/j.clineuro.2025.109030. Epub 2025 Jun 24. PMID:40570755.

6. Isikay I, Cekic E, Baylarov B, Tunc O, Hanalioglu S. Narrative review of patient-specific 3D visualization and reality technologies in skull base neurosurgery: enhancements in surgical training, planning, and navigation. Front Surg. 2024;11:1427844. doi:10.3389/fsurg.2024.1427844. PMID:39081485; PMCID:PMC11287220.

7. González-López P, Kuptsov A, Gómez-Revuelta C, et al. The Integration of 3D Virtual Reality and 3D Printing Technology as Innovative Approaches to Preoperative Planning in Neuro-Oncology. J Pers Med. 2024;14(2):187. doi: 10.3390/jpm14020187. PMID:38392620; PMCID:PMC10890029.

8. Shi Z, Peng Y, Gao X, et al. Translating high-precision mixed reality navigation from lab to operating room: design and clinical evaluation. BMC Surg. 2025;25(1):331. doi:10.1186/s12893-025-03096-0. PMID:40751242; PMCID:PMC12315425.

9. Hanalioglu S, Gurses ME, Baylarov B, Tunc O, Isikay I, Cagiltay NE, Tatar I, Berker M. Quantitative assessment and objective improvement of the accuracy of neurosurgical planning through digital patient-specific 3D models. Front Surg. 2024;11:1386091. doi:10.3389/fsurg.2024.1386091. PMID:38721022; PMCID:PMC11076751.

10. Kantak PA, Bartlett S, Chaker A, et al. Augmented Reality Registration System for Visualization of Skull Landmarks. World Neurosurg. 2024;182:369. doi:10.1016/j.wneu.2023.11.110. PMID:38013107.

11. Qi Z, Bopp MHA, Nimsky C, et al. A Novel Registration Method for a Mixed Reality Navigation System Based on a Laser Crosshair Simulator: A Technical Note. Bioengineering (Basel). 2023;10(11):1290. doi:10.3390/bioengineering10111290. PMCID:PMC10602959.

12. Begagić E, Bečulić H, Pugonja R, et al. Augmented Reality Integration in Skull Base Neurosurgery: A Systematic Review. Medicina (Kaunas). 2024;60(2):335. doi:10.3390/medicina60020335. PMID:38399622; PMCID:PMC10889940.

Recommendation

Major Revision. The authors should (1) articulate the distinct methodological contribution expected of a Lab Protocol, (2) strengthen quantitative validation with multi-case or companion data, (3) address the static-twin limitation, and (4) situate their workflow more rigorously within current mixed-reality navigation literature.

what does this mean?). If published, this will include your full peer review and any attached files.). If published, this will include your full peer review and any attached files.

Reviewer #1: No

NOTE: If reviewer comments were submitted as an attachment file, they will be attached to this email and accessible via the submission site. Please log into your account, locate the manuscript record, and check for the action link "View Attachments". If this link does not appear, there are no attachment files.]

---

## [Author Response · Author response to Decision Letter 1]

31 Oct 2025

Editor, comment #1

Please ensure that your manuscript meets PLOS ONE’s style requirements, including those for file naming. The PLOS ONE style templates can be found at (websites)

Our response #0.1

We appreciate the editor’s reminder and have checked to ensure that the manuscript complies with PLOS ONE’s formatting and file naming guidelines.

Change to Text: N/A

Editor, comment #2

In your Methods section, please provide additional information regarding the permits you obtained for the work. Please ensure you have included the full name of the authority that approved the field site access and, if no permits were required, a brief statement explaining why.

Our response #0.2

We thank the editor for the reminder. This study did not involve any fieldwork, patient recruitment, or access to clinical sites. All analyses were performed on a publicly available, fully anonymized dataset [Qi Z et al. (2024), DOI: https://doi.org/10.6084/m9.figshare.24550732.v6] released under a CC BY 4.0 license, and on 3D-printed phantoms derived from these data in a laboratory setting. Therefore, no field site access permissions or collection permits were required. We have added an explicit statement in the Methods section to clarify this.

Change to Text: [Page 5, line 130] Because the study involved no fieldwork, patient recruitment, or access to clinical or experimental sites, no field site access permissions or collection permits were required.

Editor, comment #3

We noted in your submission details that a portion of your manuscript may have been presented or published elsewhere. “Table 1 and Figure 8 are reproduced under CC-BY 4.0 license, credit:

Table 1: Amla et al. (2024), published in Journal of Digital Imaging (37(1):386–401, DOI: 10.1007/s10278-023-00912-y)

Figure 8: Qi Z et al. (2024) Head model dataset for mixed reality navigation in neurosurgical interventions for intracranial lesions. Sci Data 11: 538. doi: https://doi.org/10.1038/s41597-024-03385-y ”. Please clarify whether this [conference proceeding or publication] was peer-reviewed and formally published. If this work was previously peer-reviewed and published, in the cover letter please provide the reason that this work does not constitute dual publication and should be included in the current manuscript.

Our response #0.3

We thank the editor for the attention to this matter. Table 1 was reproduced from Amla et al., Journal of Digital Imaging (2024, peer-reviewed) under the Creative Commons Attribution 4.0 International (CC BY 4.0) license. The table summarizes publicly available comparative data on software performance and was reused without modification solely to provide contextual background on the selection of open-source software. The objectives of that review and the present study are entirely different. Figure 8 was reproduced from Qi et al., Scientific Data (2024, peer-reviewed) under the CC BY 4.0 license. It is included only to illustrate the concept of physical and virtual twins (“reference object principle”) described in the current

Methods section. The Scientific Data paper focuses on constructing a public MRN dataset for research use. In contrast, the corresponding section of this protocol explains the generation of physical and virtual twins (Steps 13–14) and their use for accuracy assessment (Step 15). Both materials are fully credited, unmodified, and reused in accordance with open-access CC BY 4.0 terms. They serve purely illustrative and methodological reference purposes and do not constitute new or overlapping results. This manuscript presents an original, comprehensive laboratory protocol that has not been published elsewhere. Therefore, inclusion of these open-licensed materials does not constitute dual publication.

Change to Text: N/A

Editor, comment #4

We note you have not yet provided a protocols.io PDF version of your protocol and/or a protocols.io DOI. When you submit your revision, please provide a PDF version of your protocol as generated by protocols.io (the file will have the protocols.io logo in the upper right corner of the first page) as a Supporting Information file. The filename should be S1 file.pdf, and you should enter “S1 File” into the Description field. Any additional protocols should be numbered S2, S3, and so on. Please also follow the instructions for Supporting Information captions [https://journals.plos.org/plosone/s/supporting-information#loc-captions]. The title in the caption should read: “Step-by-step protocol, also available on protocols.io.” Please assign your protocol a protocols.io DOI, if you have not already done so, and include the following line in the Materials and Methods section of your manuscript: “The protocol described in this peer-reviewed

article is published on protocols.io (https://dx.doi.org/10.17504/protocols.io.[...]) and is included for printing purposes as S1 File.” You should also supply the DOI in the Protocols.io DOI field of the submission form when you submit your revision.

Our response #0.4

We thank the editor for this reminder. The complete step-by-step protocol has now been formally published on protocols.io and assigned the DOI [https://dx.doi.org/10.17504/protocols.io.kqdg327p1v25/v1]. The official PDF version (with the protocols.io logo) has been uploaded as S1 File according to journal guidelines. In the “Materials and Methods” section, we have replaced the previous private link with the required citation. Change to Text: [Page 7, line 243] The protocol described in this peer-reviewed article is published on protocols.io (https://dx.doi.org/10.17504/protocols.io.kqdg327p1v25/v1) and is included for printing purposes as S1 File.

Change to Text: [Page 20, line 645] S1 File: A step-by-step protocol, also available on protocols.io (https://dx.doi.org/10.17504/protocols.io.kqdg327p1v25/v1). (PDF)

Editor, comment #5

Our response #0.5

We carefully reviewed all reviewer-suggested references. Only works directly relevant to the topic, methodology, or context of this protocol were cited.

Change to Text: N/A

Reviewer #1, comment #1

Scope and Innovation within a Lab Protocol

Even allowing for the procedural focus of a Lab Protocol, the manuscript remains largely an integration of existing 3D Slicer modules (segmentation, surface modeling, fiducial-based registration). The unique methodological advance—beyond assembling known tools—is not sufficiently articulated. Please clearly state what is genuinely novel (eg, a distinctive integration strategy, the static digital-twin validation framework, or standardized accuracy metrics) and explain why this consolidation itself represents a meaningful advance over existing Slicer tutorials or comparable platforms such as MITK or MeVisLab.

Our response #1.1

We thank the reviewer for this insightful comment. We agree that the methodological novelty of the protocol needed clearer articulation beyond the integration of existing 3D Slicer modules. In revision, we added a new subsection, “Core methodological contributions,” and expanded “Comparison with related tools and resources” and “Positioning within the literature.” These revisions clarify three key innovations: (i) standardized modular integration of imaging, segmentation, registration, and validation into a single reproducible workflow; (ii) a static digital-twin validation framework for non-contact quantitative accuracy testing; and (iii) a structured accuracy taxonomy for standardized accuracy reporting. In addition, Table 6 was added to illustrate the hierarchical three-layer architecture supporting the protocol’s clarity, reproducibility, and adaptability.

Change to Text: [Page 18, line 567] Core methodological contributions

Most prior work in this field has focused on isolated components of the workflow, without providing an end-to-end, reproducible standard for MRN development and validation. In contrast, the present protocol introduces three key methodological advances that, collectively, establish a unified foundation for MxR–based neurosurgical research and training. First, it provides standardized, modular integration of data import, segmentation, registration, and validation steps into a single, reproducible workflow that can be executed entirely within 3D Slicer. Second, it establishes a static digital twin validation framework that enables noncontact, repeatable, and quantitative testing using publicly available datasets. Third, it defines a structured accuracy taxonomy—distinguishing interpolation and extrapolation domains across virtual–to-physical and virtual–to-perceptual mappings—that standardizes accuracy reporting and facilitates objective comparison among MRN systems (see Table 3). Together, these innovations transform previously fragmented procedures into a coherent, validated methodological standard that enhances reproducibility, lowers technical barriers, and supports interdisciplinary collaboration.

Change to Text: [Page 16, line 495] Beyond the integration of software extensions, the step-by-step protocol introduces a clearly defined three-layer hierarchical structure that enhances procedural clarity, reproducibility, and modular adaptability (see Table 6 and S1 File). At the core content layer, each section, step, and sub-step is explicitly numbered. It corresponds to specific video segments in the supplementary demonstration (see S2 Appendix) and provides detailed action lists, required materials, and expected outcomes for each procedure. The supportive informational layer complements these steps with contextual annotations—such as notes, safety information, and expected results—that ensure interpretability, safety, and standardized verification. At the organizational layer, the protocol defines dependencies among both steps and extension modules, enabling their flexible recombination into six specialized pipelines optimized for different application scenarios (Fig 3). Together, this hierarchical design (summarized in Table 6) ensures transparency, facilitates learning, and enables flexible reuse and extension across diverse clinical and research contexts.

Change to Text: [Page 17, Table 6. Structural hierarchy of the step-by-step protocol based on the three-layer architecture.]

Change to Text: [Page 16, line 517] For visualization and educational applications, most reported workflows follow a similar data-processing chain that includes sequence alignment, segmentation, 3D surface reconstruction, and model optimization to enhance realism. These processes commonly involve transforming multimodal medical image coordinates into an anatomically aligned space, segmenting relevant structures, converting label maps into geometric surface models, and applying smoothing or texturing to improve visual quality [59]. Some studies further enrich the models with advanced attributes to simulate tissue deformation [59], or introduce parametric nodes to quantitatively assess trainees’ performance during simulated procedures [58]. Despite these technical advances, the underlying procedures are far from straightforward: they require close collaboration between experts in neurosurgery, anatomy, radiology, computer graphics, and engineering [59]. The resulting fragmentation of expertise and tools substantially limits efficiency and reproducibility [59]. Detailed step-by-step procedures are rarely disclosed, processing times and complexity are seldom reported, and open-source reproducibility is often lacking. The present protocol addresses these shortcomings through a modular, semi-automated design. It provides standardized parameters and explicit operational guidelines that allow for human supervision throughout while maintaining efficiency and consistency. This combination of automation and user control enhances reproducibility, reduces the burden of interdisciplinary coordination, and lowers the technical barrier for both clinicians and developers.

Reviewer #1, comment #2

Quantitative Validation and Benchmarking

The protocol commendably includes a built-in accuracy-assessment framework (TRE, FLE, displacement fields, Dice coefficient, HD95) and describes static digital-twin validation in detail. Despite these definitions, the manuscript provides no aggregated multi-case accuracy statistics, inter-observer reproducibility data, or comparative benchmarking against commercial navigation systems or other open-source pipelines. The authors need to add at least limited multi-case results or explicitly reference companion studies that report quantitative performance using this protocol to substantiate reliability and reproducibility.

Our response #1.2

We thank the reviewer for this valuable comment. In response, we have expanded the section “Use-case: MRN development, validation, and accuracy assessment” to include quantitative benchmarking data (Tables 4 and Table 5), clarify the distinction between the accuracy-assessment framework and algorithm-specific performance, and summarize empirical patterns derived from multi-case results.

Change to Text: [Page 12, line 413] To enable future studies to perform quantitative performance comparisons using this evaluation framework, the present protocol incorporates quantitative results from our group’s previously published companion studies. It provides them here as standardized benchmarks [25]. It is important to distinguish between the “accuracy assessment” itself and the “accuracy performance” of any specific system or registration algorithm. The protocol defines a generalizable evaluation scheme with related metrics in Table 3, applicable to any MRN registration paradigm. The quantitative results reported below therefore serve as reference benchmarks for assessing reproducibility and internal reliability, rather than representing fixed performance limits.

In the referenced companion studies, a novel registration method based on a laser crosshair simulator (LCS) was developed and validated using 19 patient-specific head phantoms [25]. Aggregated accuracy results across all validation cases are summarized in Table 4, and detailed case-wise data are listed in Table 5. These values constitute the first systematically validated reference benchmarks for MRN accuracy established using this protocol. Because fle cannot be directly measured or computed, and because the transformation components (TPC, TVC, and their rotational and translation subcomponents, i.e., Rx,y,z and tx,y,z), depend on the imaging coordinate system and are thus not directly comparable across cases, these parameters are not reported here.

Beyond serving as reference benchmarks, these results also reveal several characteristic empirical patterns inherent to the defined parameter framework, which can assist users in evaluating the internal consistency of their own systems:

1. Virtual–physical metrics (e.g., FRE, FN) are typically greater than their virtual–to-perceptual counterparts (fre, fn), because the registration process in the physical domain introduces additional compound errors originating from registration itself, rather than from purely geometric or perceptual factors.

2. The TRE generally exceeds both FLE and FRE. In this protocol, TRE represents a user-perceived interpolated error. Since the final MxR visualization is projected onto the user’s retina, the ability to perceive even subtle misalignments between physical and virtual structures indicates a higher-level integrated perceptual accuracy encompassing both measurement and visualization precision. In contrast, FLE and FRE capture more localized measurement deviations within single operational steps, reflecting limited sources of error and thus yielding smaller numerical values.

3. The translation magnitude t should not exceed the FN. When rotational misalignment R is small (typically less than 5°), t approximates FN, which is consistent with the mathematical definition of the transformation matrix norm.

Recognizing these empirical patterns provides practical guidance for evaluating whether new MRN implementations yield results consis

---

## [Decision Letter · Decision Letter 1]

7 Dec 2025

Dear Dr. Qi,

Thank you for submitting your manuscript to PLOS ONE. After careful consideration, we feel that it has merit but does not fully meet PLOS ONE’s publication criteria as it currently stands. Therefore, we invite you to submit a revised version of the manuscript that addresses the points raised during the review process.

If applicable, we recommend that you deposit your laboratory protocols in protocols.io to enhance the reproducibility of your results. Protocols.io assigns your protocol its own identifier (DOI) so that it can be cited independently in the future. For instructions see: https://journals.plos.org/plosone/s/submission-guidelines#loc-laboratory-protocols. Additionally, PLOS ONE offers an option for publishing peer-reviewed Lab Protocol articles, which describe protocols hosted on protocols.io. Read more information on sharing protocols at. Additionally, PLOS ONE offers an option for publishing peer-reviewed Lab Protocol articles, which describe protocols hosted on protocols.io. Read more information on sharing protocols at https://plos.org/protocols?utm_medium=editorial-email&utm_source=authorletters&utm_campaign=protocols..

We look forward to receiving your revised manuscript.

Kind regards,

Muhammad Mohsin Khan

Academic Editor

PLOS One

Journal Requirements:

Reviewers' comments:

Reviewer's Responses to Questions

**Comments to the Author**



Reviewer #1: Yes

Reviewer #2: Yes

2. Has the protocol been described in sufficient detail?

To answer this question, please click the link to protocols.io in the Materials and Methods section of the manuscript (if a link has been provided) or consult the step-by-step protocol in the Supporting Information files.

Reviewer #1: Yes

Reviewer #2: Yes

3. Does the protocol describe a validated method?

Reviewer #1: Yes

Reviewer #2: Yes

4. If the manuscript contains new data, have the authors made this data fully available?

Reviewer #1: Yes

Reviewer #2: Yes

**5. Is the article presented in an intelligible fashion and written in standard English?**

Reviewer #1: Yes

Reviewer #2: Yes

Reviewer #1: GENERAL ASSESSMENT

This manuscript presents a comprehensive, modular, and reproducible laboratory protocol for mixed reality (MxR) navigation based on the open-source platform 3D Slicer. The work effectively consolidates fragmented workflows—covering imaging, segmentation, model generation, registration, and accuracy assessment—into a single, structured protocol. The inclusion of a robust static digital twin validation framework and standardized accuracy metrics offers a meaningful methodological contribution to the field.

STRENGTHS

1. Strong methodological structure and logical workflow integration.

2. Emphasis on open-source tools and reproducible processes.

3. Robust validation framework based on a static digital twin.

4. Multi-purpose applicability (including clinical planning, education, and system validation).

5. High ethical transparency and proper use of publicly available datasets.

AREAS FOR IMPROVEMENT (Major Points)

1. The novel methodological contributions should be articulated more explicitly and distinctively, particularly in relation to existing 3D Slicer tutorials and general workflows.

2. The manuscript is lengthy and dense in certain sections; specific parts should be streamlined for clarity and conciseness.

3. The discussion of limitations should further explore the handling of dynamic digital twins or intraoperative tissue deformation.

4. The benchmark results (Tables 4 and 5) would significantly benefit from graphical visualization to facilitate clearer interpretation.

5. Detailed execution times for individual workflow steps should be included to strengthen the protocol's reproducibility.

MINOR REVISION SUGGESTIONS

• Improve consistency in figure and table captions and numbering.

• Consolidate repetitive statements within the Introduction and Discussion sections.

• Simplify certain explanations of the accuracy metrics and accompanying workflow diagrams.

• Ensure uniform reference formatting across the entire manuscript.

This manuscript is well-developed, adheres to PLOS ONE protocol standards, and successfully addresses an important methodological gap in mixed reality–based neurosurgical workflow development. The recommended revisions are primarily focused on enhancing clarity, conciseness, and improving the visibility of the key methodological innovations.

Recommendation: Minor Revision

Reviewer #2: I would like to express my sincere gratitude for the opportunity to review this manuscript.

This work comprehensively summarizes the authors’ ongoing research on Mixed Reality Navigation (MRN) and provides substantial contributions as a knowledge base for the field.

The workflow based on 3D Slicer is clearly structured and applicable to users ranging from beginners to advanced practitioners. The manuscript offers valuable and practical information that enables readers to efficiently understand and reproduce the operational steps.

In particular, the inclusion of instructional videos is highly commendable; the demonstrations greatly enhance the clarity and reproducibility of the protocol.

Although the current protocol may not represent a fully perfected workflow, its publication will likely facilitate broader adoption of 3D Slicer–based MRN approaches. This wider use could, in turn, promote further technical developments and improvements in usability—for example, integration into a unified module or dedicated application—which may significantly benefit the community.

At the same time, I would like to suggest several points for improvement prior to publication.

1. Limited novelty and redundancy of explanations

Much of the content is based on existing 3D Slicer functionality and on the authors’ previously published work, making the novelty of this manuscript inherently limited.

In addition, detailed explanations added in response to the first-round reviewers' comments have resulted in redundancy, particularly in the latter sections describing MRN accuracy evaluation. Many of these descriptions rely heavily on previously published data and could be summarized more concisely, with details referred to the original publications.

Conversely, genuinely new data—such as the companion analysis results presented in Tables 4 and 5—should be highlighted more clearly.

2. Insufficient presentation of Virtual–Physical alignment

Given that the protocol aims to evaluate Virtual–Physical consistency, the figures currently presented are insufficient. Most of the visual examples rely solely on Virtual models, and it is not clear how Virtual contents are aligned with actual Physical objects.

For scientific credibility, at least one figure should demonstrate that part of the Virtual object designed in the digital space was 3D-printed into a Physical twin, and that the Virtual contents were correctly projected and aligned onto this Physical model using the MRN system.

3. Unclear rationale for the use of the Greek sculpture in Figure 4

Figure 4 appears to show Virtual organ models or image overlays projected onto a Greek sculpture. It is unclear why such an example is used instead of projecting onto a 3D-printed model derived from the patient’s own cranial anatomy.

Given that this study focuses on MRN accuracy evaluation using Digital Twins, the use of a Greek sculpture is difficult to interpret in this context. Presenting an example using a patient-specific Physical model would be far more appropriate and intuitive for readers.

Overall, I believe that addressing these concerns will further enhance the clarity, scientific rigor, and practical value of the manuscript.

what does this mean?). If published, this will include your full peer review and any attached files.). If published, this will include your full peer review and any attached files.

Reviewer #1: No

Reviewer #2: No

---

## [Author Response · Author response to Decision Letter 2]

13 Dec 2025

Reviewer #1, comment #1

The novel methodological contributions should be articulated more explicitly and distinctively, particularly in relation to existing 3D Slicer tutorials and general workflows.

Our response #1.1

Thank you for this suggestion. We revised the manuscript to make the protocol’s methodological novelty explicit and clearly distinguish it from existing 3D Slicer tutorials and general workflows. Specifically, we (i) added an explicit contrast explaining why current Slicer resources are typically module-centric and do not provide an MRN-specific end-to-end pipeline, and (ii) summarized the protocol’s distinct methodological contributions (end-to-end MRN workflow integration, static digital-twin validation, and a unified accuracy taxonomy), including the coupling to step-level time estimates and standardized benchmarks.

Change to Text: [Page 2–3, lines 55–74] To address these challenges, this protocol, informed by multiple peer-reviewed studies [23, 25, 39], proposes a structured, modular protocol for the open-source

platform 3D Slicer, specifically designed for MRN implementation and evaluation in neurosurgical contexts. Existing 3D Slicer tutorials and community resources are largely module-centric and typically cover single tasks (e.g., segmentation or visualization), offering limited guidance on how to connect these components into an MRN-specific pipeline that supports registration, twin-based validation, and quantitative benchmarking. Importantly, this protocol couples this workflow with step-level time estimates and standardized benchmark results or visualizations, enabling direct replication and quantitative comparison across MRN implementations. Rather than introducing yet another isolated example workflow, the protocol: (i) unifies image import, multimodal preprocessing, anatomical and functional modelling, registration preparation, and accuracy evaluation into a single, end-to-end MRN workflow; (ii) embeds a static digital-twin validation paradigm that enables image-derived, non-contact, and repeatable accuracy measurements without external CAD software or custom scripts; and (iii) formalizes an accuracy framework that distinguishes interpolation from extrapolation behaviour and separates virtual-to-physical from virtual-to-perceptual error domains. Together, these elements provide a reproducible methodological standard for MRN development that can be applied to surgical planning, education, and system validation.

Change to Text: [Page 13, lines 383–385] While 3D Slicer documentation and tutorials are extensive, they are typically feature- or module-centric and rarely describe how to assemble a complete MRN workflow end-to-end.

Reviewer #1, comment #2

The manuscript is lengthy and dense in certain sections; specific parts should be streamlined for clarity and conciseness.

Our response #1.2

Thank you for this comment. We streamlined several dense sections to improve readability. In particular, we condensed the main-text description of the MRN accuracy evaluation into a concise conceptual summary. We moved detailed metric definitions and extended examples to S4 Appendix (Change to Text: [Page 7, lines 226–237]). We also rewrote sections (Comparison with related tools and sources; Positioning within the literature) to be more concise and reduced repetition in the later narrative sections (Change to Text: [Pages 11–13, lines 369–421]). In addition, we removed the Table “Structural hierarchy of the step-by-step protocol based on the three-layer architecture” and the Figure “Overview of the validation reference object principle.”

Reviewer #1, comment #3

The discussion of limitations should further explore the handling of dynamic digital twins or intraoperative tissue deformation.

Our response #1.3

Thank you for highlighting this important point. We expanded the “Limitations and future directions” section to more explicitly discuss why the current protocol focuses on static digital twins as a reproducible, rigid baseline, and to outline how the framework could be extended toward dynamic digital twins and intraoperative deformation handling. The revision now (i) clarifies the intended scope (phantom-based validation and initial registration assessment), (ii) summarizes major sources of intraoperative deformation (e.g., brain shift mechanisms), and (iii) describes compatible pathways for future extensions (e.g., intraoperative surface acquisition, non-rigid registration, biomechanical/FEM modeling, and AI-based surface reconstruction), emphasizing that the presented accuracy taxonomy and modular design can support time-varying evaluations. Change to Text: [Page 14, lines 431–461] Secondly, the current protocol is limited to static digital twins, which serve as fixed geometric references and do not track real-time deformations [65]. This static design intentionally reflects the intended scope of the present work: to establish a reproducible, rigid baseline for evaluating MRN registration accuracy under controlled conditions. Such static twins remain indispensable, as any future dynamic or deformable MRN system must first demonstrate reliable performance in a stable and repeatable reference space before more complex, time-varying validations become meaningful. However, static twins cannot account for intraoperative tissue deformation. Accordingly, the present protocol is intended as a rigid baseline primarily for phantom-based validation and initial registration assessment (i.e., before substantial intraoperative deformation), rather than for modeling post-dural-opening brain shift or other time-varying tissue dynamics. At the cranial level, subtle skull deflection introduced by fixation pins or clamps may cause small but relevant rigid offsets before dural opening [1]. At deeper anatomical layers, dynamic intracranial deformation—including gravity-induced brain shift, cerebrospinal fluid egress, retraction effects, and resection-cavity collapse—cannot be represented by static replicas [1]. Progress toward dynamic digital twins will require coupling intraoperative measurements with computational deformation models. Several technological pathways are compatible with, and may extend, the protocol presented here. Real-time surface acquisition from intraoperative ultrasound or depth cameras can provide time-varying point clouds suitable for non-rigid registration using existing tools such as Elastix B-spline transforms [65].

Modules such as Segment Mesher and SlicerBiomech enable the creation of patient-specific tetrahedral meshes for finite-element solvers, whose predicted deformation fields can be reimported to warp preoperative volumes [66,67]. AI-based monocular depth-estimation approaches offer an additional low-cost means for reconstructing intraoperative cortical surfaces when advanced imaging modalities are unavailable [58,68]. Together, these avenues outline a coherent progression toward dynamic digital twins based on (i) real-time surface updates, (ii) biomechanical or data-driven deformation modeling, and (iii) continuous refinement of virtual-to-physical alignment. Importantly, the accuracy taxonomy and modular design of the present protocol are compatible with future extensions toward time-varying, 4D evaluations.

Reviewer #1, comment #4

The benchmark results (Tables 4 and 5) would significantly benefit from graphical visualization to facilitate clearer interpretation.

Our response #1.4

Thank you for this suggestion. We now present the benchmark accuracy metrics as boxplots with jittered scatter points. (Change to Text: [Page 11, Added Fig 8] “Benchmark accuracy metrics across 19 cases

and 21 lesions”)

Reviewer #1, comment #5

Detailed execution times for individual workflow steps should be included to strengthen the protocol’s reproducibility.

Our response #1.5

Thank you for this recommendation. To further strengthen reproducibility and practical usability, we added a step-level execution-time summary table. The new table reports typical durations for each major step (Steps 1–15) and clarifies that these values were derived from the recorded demonstration videos and therefore represent upper-bound estimates, while actual routine execution may be shorter depending on case complexity and user experience. (Change to Text: [Added Table 3] “Typical execution time for each major step of the protocol.”)

Reviewer #1, comment #6

MINOR REVISION SUGGESTIONS • Improve consistency in figure and table captions and numbering. • Consolidate repetitive statements within the Introduction and Discussion sections. • Simplify certain

explanations of the accuracy metrics and accompanying workflow diagrams. • Ensure uniform reference formatting across the entire manuscript.

Our response #1.6

Thank you for these helpful minor suggestions. We performed a full consistency and readability pass across the manuscript. Specifically, we (i) standardized figure and table caption style and ensured that all

numbering and in-text cross-references are consistent; (ii) consolidated repetitive statements, particularly in the narrative and discussion-style sections, to reduce redundancy while preserving key messages; (iii) simplified the main-text explanations of the accuracy metrics and workflow by keeping a high-level conceptual description in the manuscript and moving detailed metric definitions and extended examples to Supporting Information; and (iv) harmonized reference formatting throughout the bibliography (including DOI formatting and minor typographic corrections).

Reviewer #2, comment #1

Limited novelty and redundancy of explanations Much of the content is based on existing 3D Slicer functionality and on the authors’ previously published work, making the novelty of this manuscript

inherently limited. In addition, detailed explanations added in response to the first-round reviewers’ comments have resulted in redundancy, particularly in the latter sections describing MRN accuracy evaluation. Many of these descriptions rely heavily on previously published data and could be summarized more concisely, with details referred to the original publications. Conversely, genuinely new data—such as the companion analysis results presented in Tables 4 and 5—should be highlighted more clearly.

Our response #2.1

Thank you for this constructive critique. We revised to clarify the manuscript’s incremental methodological contribution beyond existing 3D Slicer functionality and our prior studies, emphasizing that the novelty lies in formalizing an MRN-specific, end-to-end, reusable protocol standard (with modular pipelines, transparent step-level time reporting, and a reproducible validation/benchmarking framework), rather than introducing isolated tool demonstrations. In addition, we streamlined the latter accuracy-evaluation narrative by moving detailed metric definitions and extended examples to the S4 Appendix (Change to Text: [Page 7, lines 226–237]), keeping a concise conceptual description in the main text (Change to Text: [Pages 11–13, lines 369–421]), and by explicitly positioning Tables 4–5 as protocol-linked reference benchmarks and adding a graphical summary of their distributions. (Change to Text: [Page 11, Added Fig 8] “Benchmark accuracy metrics across 19 cases and 21 lesions”)

Change to Text: [Page 14, lines 431–461] Secondly, the current protocol is limited to static digital twins, which serve as fixed geometric references and do not track real-time deformations [65]. This static design intentionally reflects the intended scope of the present work: to establish a reproducible, rigid baseline for evaluating MRN registration accuracy under controlled conditions. Such static twins remain indispensable, as any future dynamic or deformable MRN system must first demonstrate reliable performance in a stable and repeatable reference space before more complex, time-varying validations become meaningful. However, static twins cannot account for intraoperative tissue deformation. Accordingly, the present protocol is intended as a rigid baseline primarily for phantom-based validation and initial registration assessment (i.e., before substantial intraoperative deformation), rather than for modeling post-dural-opening brain shift or other time-varying tissue dynamics. At the cranial level, subtle skull deflection introduced by fixation pins or clamps may cause small but relevant rigid offsets before dural opening [1]. At deeper anatomical layers, dynamic intracranial deformation—including gravity-induced brain shift, cerebrospinal fluid egress, retraction effects, and resection-cavity collapse—cannot be represented by static replicas [1]. Progress toward dynamic digital twins will require coupling intraoperative measurements with computational deformation models. Several technological pathways are compatible with, and may extend, the protocol presented here. Real-time surface acquisition from intraoperative ultrasound or depth cameras can provide time-varying point clouds suitable for non-rigid registration using existing tools such as Elastix B-spline transforms [65]. Modules such as Segment Mesher and SlicerBiomech enable the creation of patient-specific tetrahedral meshes for finite-element solvers, whose predicted deformation fields can be reimported to warp preoperative volumes [66,67]. AI-based monocular depth-estimation approaches offer an additional low-cost means for reconstructing intraoperative cortical surfaces when advanced imaging modalities are unavailable [58,68]. Together, these avenues outline a coherent progression toward dynamic digital twins based on (i) real-time surface updates, (ii) biomechanical or data-driven deformation modeling, and (iii) continuous refinement of virtual-to-physical alignment. Importantly, the accuracy taxonomy and modular design of the present protocol are compatible with future extensions toward time-varying, 4D evaluations.

Reviewer #2, comment #2

Insufficient presentation of Virtual–Physical alignment Given that the protocol aims to evaluate Virtual–Physical consistency, the figures currently presented are insufficient. Most of the visual examples rely solely on Virtual models, and it is not clear how Virtual contents are aligned with actual Physical objects. For scientific credibility, at least one figure should demonstrate that part of the Virtual object designed in the digital space was 3D-printed into a Physical twin, and that the Virtual contents were correctly projected and aligned onto this Physical model using the MRN system.

Our response #2.2

Thank you for this important point. We agree that demonstrating virtual-to-physical alignment is essential for scientific credibility in an MRN protocol. We therefore added a dedicated figure that shows mixed-reality overlays captured by the MRN system after registration, using two patient-specific 3D-printed head phantoms (physical twins) together with their corresponding virtual content. (Change to Text: [Page 11, Added Fig 7] “Demonstration of virtual-to-physical alignment.”)

Reviewer #2, comment #3

Unclear rationale for the use of the Greek sculpture in Fig 4 Fig 4 appears to show Virtual organ models or image overlays projected onto a Greek sculpture. It is unclear why such an example is used instead of projecting onto a 3D-printed model derived from the patient’s own cranial anatomy. Given that this study focuses on MRN accuracy evaluation using Digital Twins, the use of a Greek sculpture is difficult to interpret in this context. Presenting an example using a patient-specific Physical model would be far more appropriate and intuitive for readers.

Our response #2.3

Thank you for pointing this out. We agree that the Greek sculpture example can be confusing bere. Hence, we remove the Greek sculpture figure panels (Change to Text: [Page 9, removed panels E and F in Fig 3] “Representative outputs of segmentation and visualization from pipelines”) and added a dedicated figure that shows MxR overlays using two patient-specific 3D-printed head phantoms. (Change to Text: [Page 11, Added Fig 7] “Demonstration of virtual-to-physical alignment.”)

---

## [Decision Letter · Decision Letter 2]

16 Feb 2026

Comprehensive protocol for mixed reality visualization and navigation using 3D Slicer

PONE-D-25-39628R2

Dear Dr. Qi,

We’re pleased to inform you that your manuscript has been judged scientifically suitable for publication and will be formally accepted for publication once it meets all outstanding technical requirements.

An invoice will be generated when your article is formally accepted. Please note, if your institution has a publishing partnership with PLOS and your article meets the relevant criteria, all or part of your publication costs will be covered. Please make sure your user information is up-to-date by logging into Editorial Manager at Editorial Manager®  and clicking the ‘Update My Information' link at the top of the page. For questions related to billing, please contact  and clicking the ‘Update My Information' link at the top of the page. For questions related to billing, please contact billing support..

Kind regards,

Mohammad Mofatteh, PhD, MPH, MSc, PGCert, BSc (Hons), MB BCh (c)

Academic Editor

PLOS One

https://scholar.google.com/citations?user=U_uB130AAAAJ&hl=en

Additional Editor Comments (optional):

Thank you for resubmitting your manuscript. The authors have addressed the comments well.

Reviewers' comments:

Reviewer's Responses to Questions

**Comments to the Author**



Reviewer #1: Yes

Reviewer #3: Yes

Reviewer #4: Yes

2. Has the protocol been described in sufficient detail?

To answer this question, please click the link to protocols.io in the Materials and Methods section of the manuscript (if a link has been provided) or consult the step-by-step protocol in the Supporting Information files.

Reviewer #1: Yes

Reviewer #3: Yes

Reviewer #4: Yes

3. Does the protocol describe a validated method?

Reviewer #1: Yes

Reviewer #3: Yes

Reviewer #4: No

4. If the manuscript contains new data, have the authors made this data fully available?

Reviewer #1: Yes

Reviewer #3: Yes

Reviewer #4: N/A

**5. Is the article presented in an intelligible fashion and written in standard English?**

Reviewer #1: Yes

Reviewer #3: Yes

Reviewer #4: Yes

Reviewer #1: The authors have satisfactorily addressed the minor revision suggestions. The manuscript has improved in terms of clarity, consistency, and overall presentation. I have no further comments and consider the manuscript suitable for publication in its current form.

Reviewer #3: The authors present a protocol informed by prior literature and existing tutorials on the capabilities of the open-source DICOM viewer 3D Slicer for mixed-reality visualization and navigation. The protocol outlines fundamental steps for image upload, anonymization, and anatomical segmentation for beginners, and progresses to more advanced functions, including the generation of three-dimensional (3D) surface models suitable for 3D printing, as well as for virtual and mixed-reality applications (VR and MR). Additionally, the protocol describes key procedures for the parameterization of fiducial landmarks to enable accurate virtual-to-physical registration for navigation.

I have read with great interest the reviewers’ previous comments and the authors’ detailed responses. The prior two revision rounds have substantially improved the quality and clarity of the manuscript. In my view, this protocol serves as a valuable vade mecum for colleagues approaching MRI- and CT-based 3D model visualization and navigation using this software, particularly for those adopting 3D Slicer for the first time. While the manuscript does not introduce conceptual novelty, its strength lies in the structured synthesis of essential steps for creating, visualizing, and navigating image-derived 3D patient representations. On this basis, I believe the protocol will be highly useful to the readership. I found the step-by-step guide provided in the supplementary material especially instructive, supported by clear tutorial videos that effectively orient new users to key software workflows and learning milestones.

Overall, I consider this protocol to be a meaningful and practical contribution that will facilitate broader adoption and correct application of 3D Slicer in clinical 3D visualization and navigation.

Minor revisions:

- Please restate the specifics of the computer you used to run the different steps of the protocol in the legend of Table 2.

- I suggest reducing the introduction to no more than 8-10 lines, removing duplicated gap/complexity/resource-limited narrative.

Reviewer #4: Following the authors’ revisions, the work now clearly positions itself as a methodological and operational standard, rather than as an algorithmic novelty paper.

The protocol is technically sound, reproducible, and well aligned with the scope of PLOS ONE, particularly given its emphasis on open-source workflows, transparency, and validation using static digital twins. The authors have responded thoroughly to prior reviewer comments, significantly improving clarity, rigor, and presentation.

I recommend acceptance of the manuscript, subject only to minor, optional refinements outlined below:

- Although the authors clearly state that the protocol is intended as a labratory and phantom-based baseline, the discussion could more explicitly articulate a concrete pathway toward preclinical experimentation, such as cadaveric studies, animal models, or controlled operating-room simulations prior to clinical translation. Even a short paragraph clarifying how the proposed protocol could be embedded into a staged valiadtion pipeline (phantom → cadaver → preclinical → clinical) would further enhance its translational relevance. In this context the authors may consider citing recent preclinical evaluations of neuronavigation systems that report quantitative accuracy on patient-specific phantoms, such as Carbone et al., Scientific Reports (2025) - https://www.nature.com/articles/s41598-025-14555-2. This would further contextualize the proposed protocol within current efforts toward rigorous preclinical validation of AR/MR navigation systems.

what does this mean?). If published, this will include your full peer review and any attached files.). If published, this will include your full peer review and any attached files.

Reviewer #1: No

Reviewer #3: No

Reviewer #4: No

---

## [Editor Report · Acceptance letter]

PONE-D-25-39628R2

PLOS One

Dear Dr. Qi,

I'm pleased to inform you that your manuscript has been deemed suitable for publication in PLOS One. Congratulations! Your manuscript is now being handed over to our production team.

Kind regards,

on behalf of

Dr. Mohammad Mofatteh

Academic Editor

PLOS One